# LANG-PRUNE: UNLOCKING FAIR AND POWERFUL PRUNING FOR MULTILINGUAL LARGE LANGUAGE MODELS

## ABSTRACT

Multilingual large language models (LLMs) are essential for cross-lingual applications, yet pruning them using mixed-language calibration can induce cross-lingual interference, disproportionately affecting certain languages. We introduce *Lang-Prune*, a drop-in, language-aware extension to structured pruning that computes per-language importance on small calibration sets and aggregates it to protect units critical to any language. Evaluated on `aya-expanse-8b` across nine languages and multiple sparsity levels, Lang-Prune consistently improves both average and worst-case performance. At 70% sparsity, it reduces average perplexity from 188.49 (original pruning method) to 70.85, surpassing the monolingual baseline (83.08) while lowering the worst-language error. Interpretability analyses reveal higher retention of language-specific capacity (81% vs 66%). Ablations demonstrate robustness across model types (e.g., `Qwen3-8B`), improved post-training headroom, and strong transfer to out-of-distribution languages. Lang-Prune is compute-efficient and deployment-friendly, requiring only modifications to importance estimation and aggregation while preserving LLM-Pruner's coupled-structure mechanics.

## 1 INTRODUCTION

With the rapid advancement of large language models (LLMs) OpenAI et al. (2024); Touvron et al. (2023), people worldwide are increasingly benefiting from this technology. To further broaden its impact, researchers have devoted substantial effort to collecting low-resource language data and developing multilingual LLMs with larger parameter scales and stronger capabilities Chen et al. (2023); Yang et al. (2025). However, the massive size of these models imposes heavy computational and memory demands, restricting their deployment in resource-constrained environments such as mobile devices and causing significant latency in client–server interactions. Model pruning has emerged as a practical solution to alleviate these challenges by removing redundant parameters while striving to maintain performance, particularly when adapting an existing, well-aligned model to resource-constrained deployments Wang et al. (2020); Xia et al. (2022); Muralidharan et al. (2024); Xia et al.; Kong et al. (2025).

Currently, most existing pruning methods are evaluated primarily on monolingual or high-resource languages, often neglecting the cross-lingual variability inherent in multilingual LLMs Sun et al. (2024); Frantar & Alistarh (2023). Our pilot study reveals a central obstacle: cross-lingual interference. When pruning with mixed-language calibration, decisions optimized for average case disproportionately harm certain languages. Using the *interference factor (IF)*, defined as the ratio of mixed vs monolingual perplexity per language, we find consistent degradation across nine languages for both structured (LLM-Pruner Ma et al. (2023)) and unstructured (SparseGPT Frantar & Alistarh (2023)) baselines, with stronger effects under structured coupling.

We introduce *Lang-Prune*, a language-aware extension to LLM-Pruner that estimates importance per language on small calibration sets and aggregates these scores to protect units critical to any language. Lang-Prune preserves LLM-Pruner's coupled-structure mechanics and sparsity schedules, modifying only the scoring and aggregation process. Experiments across nine languages and multiple sparsity ratios show that Lang-Prune consistently improves both average and worst-case

performance, often surpassing monolingual pruning even when using multilingual calibration. Interpretability analyses indicate that it better retains language-specific capacity. Extensive ablation studies further demonstrate that *Lang-Prune* (1) **adapts to multiple model types**, (2) **preserves the potential of pruned LLMs for post-training**, and (3) **exhibits strong transfer to out-of-distribution languages**. Overall, this work presents a practical framework for multilingual LLM compression, enabling efficient deployment without compromising language coverage.

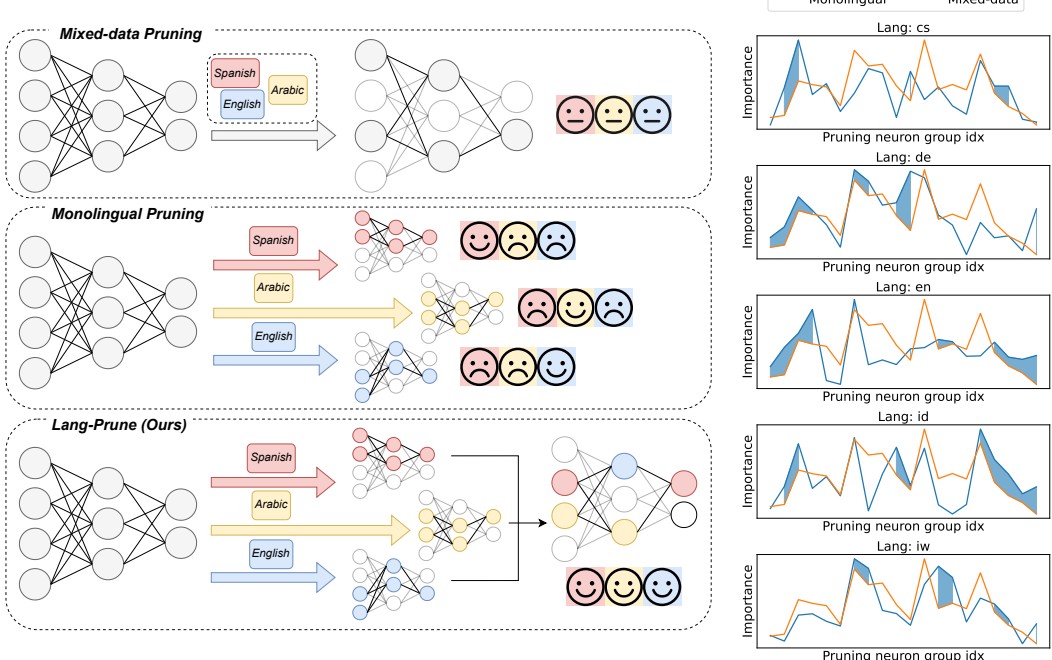

Figure 1: **Left:** Illustration of three pruning strategies—*Mixed-data Pruning*, *Monolingual Pruning*, and *Lang-Prune*. **Right:** Language-specific neuron-group importance under monolingual vs. mixed-data calibration. Shaded areas indicate neurons important for a single language but less salient under multilingual pruning.

**Contributions:** This paper makes the following contributions: (1) We present *Lang-Prune*, a language-aware pruning framework that computes per-language importance and aggregates to mitigate cross-lingual interference while preserving deployment-friendly structure. (2) On nine typologically diverse languages, Lang-Prune improves both average and worst-case performance across sparsity ratios, often outperforming monolingual pruning under multilingual calibration. (3) Extensive ablations show generalization across model types, compatibility with post-training, and strong zero-shot transfer to out-of-distribution languages. (4) Interpretability analyses reveal that Lang-Prune retains language-specific neuron groups, offering insight into multilingual capacity preservation during compression.

## 2 LANG-PRUNE: A MULTILINGUAL PRUNING FRAMEWORK

### 2.1 PILOT STUDY: CROSS-LINGUAL INTERFERENCE ON PRUNING

We conduct a pilot study to quantify how existing pruning methods behave in multilingual settings. We evaluate a structured pruning method, LLM-Pruner Ma et al. (2023), and an unstructured pruning method, SparseGPT Frantar & Alistarh (2023), on `aya-expanse-8b` Dang et al. (2024). The experimental settings match those in the main experiments (see Section 3). In the Mixed-data Pruning setting, we pool all nine languages (900 sequences total, uniformly sampled per language); in the Monolingual Pruning setting, pruning is performed separately for each language using 100 sequences, yielding nine language-specific pruned checkpoints. To quantify cross-lingual interference,

we report the Cross-lingual Interference Factor (IF) defined as $\text{IF}(l) = \text{PPL}_{\text{mixed}}(l)/\text{PPL}_{\text{mono}}(l)$, where values above 1 indicate performance degradation due to mixed-language calibration.

Table 1: Cross-lingual Interference Factor (IF) for multilingual pruning. Values greater than 1 indicate performance degradation due to cross-lingual interference.

| *LLM-Pruner under 70% Sparsity* | | | | | | | | | | |
|---|---|---|---|---|---|---|---|---|---|---|
| *PPL score ↓* | ar | cs | de | en | es | id | iw | ru | zh | avg. |
| Monolingual | 42.71 | 81.99 | 82.82 | 236.44 | 85.19 | 77.52 | 46.87 | 57.73 | 36.47 | 83.08 |
| Mixed-data | 80.99 | 164.12 | 245.65 | 378.39 | 227.26 | 173.63 | 71.97 | 127.51 | 226.88 | 188.49 |
| IF score | 1.90× | 2.00× | 2.97× | 1.60× | 2.67× | 2.24× | 1.54× | 2.21× | 6.22× | 2.27× |
| Base Model | 8.10 | 10.46 | 10.66 | 10.50 | 9.58 | 12.77 | 11.19 | 11.13 | 10.27 | 10.52 |
| *SparseGPT under 70% Sparsity* | | | | | | | | | | |
| *PPL score ↓* | ar | cs | de | en | es | id | iw | ru | zh | avg. |
| Monolingual | 18.75 | 27.57 | 24.69 | 31.04 | 26.83 | 25.97 | 21.72 | 22.51 | 18.96 | 25.94 |
| Mixed-data | 27.72 | 38.46 | 32.51 | 40.53 | 32.57 | 34.26 | 31.16 | 30.97 | 29.72 | 34.84 |
| IF score | 1.48× | 1.39× | 1.32× | 1.31× | 1.21× | 1.32× | 1.43× | 1.38× | 1.57× | 1.38× |
| Base Model | 8.10 | 10.46 | 10.66 | 10.50 | 9.58 | 12.77 | 11.19 | 11.13 | 10.27 | 10.52 |

As shown in Table 1, Mixed-data Pruning consistently underperforms Monolingual Pruning across all nine languages and both pruning paradigms. For structured pruning (LLM-Pruner), the average IF is 2.27×, with particularly severe degradation for zh (6.22×), de (2.97×), and es (2.67×). Unstructured pruning (SparseGPT) exhibits a smaller but systematic effect (average IF 1.38×; maximum 1.57× for zh). Notably, the mixed-language setting uses a larger aggregate calibration budget (900 vs. 100 sequences) yet still underperforms language-specific pruning, making these results conservative.

To investigate the source of this degradation, Figure 1 visualizes LLM-Pruner saliency across neuron groups for five languages. Under monolingual calibration, importance profiles show sharp, language-specific peaks that are well-aligned with each language's critical structures; under multilingual mixed-data calibration, these peaks become **attenuated, misaligned, or replaced by peaks arising from multilingual interference**. This mismatch causes language-critical groups to appear less salient and more likely to be pruned globally—a phenomenon we term ***cross-lingual interference in pruning***: *structures vital for certain languages may seem unimportant when pooled with others and thus get pruned.* This effect is particularly pronounced in structured pruning, where coupled units concentrate language-specific capacity, amplifying interference.

## 2.2 OVERVIEW OF LANG-PRUNE

To address cross-lingual interference and achieve balanced pruning across languages, we introduce **Lang-Prune**, a multilingual-aware extension to LLM-Pruner guided by *language-specific importance scores*. As shown in the left panel of Figure 1, unlike conventional approaches that estimate neuron importance from a single calibration set, Lang-Prune computes importance scores per language and aggregates them using fairness-aware policies, ensuring that structures critical to any language are preserved during pruning.

This design is motivated by evidence that multilingual LLMs contain both universal and language-specific mechanisms: some units encode language-agnostic patterns, while others capture script- or morphology-dependent features Singh et al. (2019); Liu et al. (2019); Conneau et al. (2020); Zheng et al. (2025). Building on these insights, we extend LLM-Pruner's structured pruning pipeline to explicitly respect per-language signals. The process consists of three stages: (1) coupled structure discovery; (2) language-aware importance estimation; (3) multilingual importance scores aggregation and pruning.

### 2.2.1 COUPLED STRUCTURE DISCOVERY IN LLM-PRUNER

LLM-Pruner Ma et al. (2023) discovers valid, shape-consistent pruning units by constructing a dependency graph over Transformer components and grouping parameters that must be pruned jointly (coupled structures). This enables structured pruning of attention heads and feed-forward (MLP) channels without breaking tensor shapes or deployment compatibility.

For example, in a feed-forward block, pruning the $j$-th hidden channel requires removing the $j$-th column of the input projection and the $j$-th row of the output projection simultaneously. LLM-Pruner

treats this pair as a single coupled structure, denoted $\mathcal{C}_j$, and assigns it a score based on parameter magnitude and activation statistics collected on calibration data. This coupled view generalizes across modules and forms the backbone of Lang-Prune.

Lang-Prune extends this framework by modifying how scores are computed and aggregated: instead of using global metrics, scores are calculated per language and then combined to guide multilingual pruning. As our ablation study in Appendix A.7 demonstrates on Wanda, preserving per-language importance at the level of functional components (e.g., attention heads, MLP channels) is far more impactful than focusing on isolated weights. Therefore, **the coupled structure discovery mechanism inherited from LLM-Pruner is essential for Lang-Prune**, as it ensures that language-specific importance scores meaningfully influence pruning decisions while maintaining model deployability.

### 2.2.2 LANGUAGE-AWARE IMPORTANCE ESTIMATION

Given the set of coupled structures $\{\mathcal{C}_j\}$, we estimate their importance independently for each language using a loss-sensitivity criterion computed on language-specific calibration subsets. Let $\mathcal{L}$ denote the set of languages and $\mathcal{D}_\ell$ the calibration data for language $\ell$. For a coupled structure $\mathcal{C}_j$, let $\mathcal{G}(\mathcal{C}_j)$ be the collection of learnable tensors (or parameter vectors) that constitute the structure (e.g., paired columns/rows in coupled projections). Following the Taylor-based importance estimator in LLM-Pruner Ma et al. (2023), we measure the first-order contribution of each parameter to the next-token prediction loss.

Concretely, let $L(x)$ denote the token-averaged negative log-likelihood (next-token prediction) for input $x$. For a scalar parameter $w \in \mathcal{C}_j$, the per-parameter, per-language importance is

$$s_\ell(w) \;=\; \mathbb{E}_{x \sim \mathcal{D}_\ell} \left| \frac{\partial L(x)}{\partial w} \cdot w \right|, \tag{1}$$

where the absolute value ensures non-negativity and robustness to sign cancellations. The expectation is approximated by the empirical mean over the calibration subset, with the model in evaluation mode (no dropout) and gradients computed at the current weights without updating them.

When treating an entire tensor $W \in \mathcal{G}(\mathcal{C}_j)$ as a unit, we use the vectorized form

$$s_\ell(W) \;=\; \mathbb{E}_{x \sim \mathcal{D}_\ell} \left| \left\langle \frac{\partial L(x)}{\partial W}, W \right\rangle \right|, \quad \text{with } \langle A, B \rangle \;=\; \mathrm{tr}(A^\top B), \tag{2}$$

i.e., the absolute value of the Frobenius inner product between the gradient and the parameter tensor.

We aggregate the parameter- or tensor-level scores into a structure-level importance using a group aggregator $\mathcal{A}$ over all elements of $\mathcal{G}(\mathcal{C}_j)$. By default, we adopt the `sum` aggregator:

$$I_\ell(\mathcal{C}_j) \;=\; \sum_{u \in \mathcal{G}(\mathcal{C}_j)} s_\ell(u), \tag{3}$$

where $u$ denotes either a scalar weight (using Eq. 1) or a tensor (using Eq. 2). Other aggregators (e.g., `max`, `mean`, `first`) are supported and evaluated in ablations in Appendix A.2.

To make scores comparable across languages, we apply per-language min–max normalization over all coupled structures:

$$\tilde{I}_\ell(\mathcal{C}_j) \;=\; \frac{I_\ell(\mathcal{C}_j) - \min_k I_\ell(\mathcal{C}_k)}{\max_k I_\ell(\mathcal{C}_k) - \min_k I_\ell(\mathcal{C}_k) + \varepsilon}, \tag{4}$$

where the extrema are computed over all coupled structures indexed by $k$, and a small $\varepsilon$ (e.g., $10^{-12}$) ensures numerical stability. In the rare degenerate case where $\max_k I_\ell(\mathcal{C}_k) \approx \min_k I_\ell(\mathcal{C}_k)$, this normalization yields $\tilde{I}_\ell(\mathcal{C}_j) \approx 0$ for all $j$, effectively indicating no preference among structures for language $\ell$. This language-wise normalization mitigates the effects of tokenization, script differences, and frequency imbalance, while preserving the importance ranking within each language.

### 2.2.3 Pruning LLMs with Multilingual Importance Scores

Lang-Prune merges per-language importances into a single score that guards against worst-case language degradation. Given $\{\tilde{I}_\ell(\mathcal{C}_j)\}_{\ell\in\mathcal{L}}$, we use the **Max** aggregator:

$$I_{\max}(\mathcal{C}_j) = \max_{\ell\in\mathcal{L}} \tilde{I}_\ell(\mathcal{C}_j). \tag{5}$$

This "any-language" criterion preserves structures that are important for at least one language, directly countering the dilution effect observed in mixed-language calibration. For comparison in ablations, we also consider the **Mean** and **Min** aggregators:

$$I_{\text{mean}}(\mathcal{C}_j) = \frac{1}{|\mathcal{L}|} \sum_{\ell\in\mathcal{L}} \tilde{I}_\ell(\mathcal{C}_j), \quad I_{\min}(\mathcal{C}_j) = \min_{\ell\in\mathcal{L}} \tilde{I}_\ell(\mathcal{C}_j). \tag{6}$$

Structures are ranked by $I_{\max}$ (our default) and pruned (lowest first) until the target sparsity is reached, strictly respecting LLM-Pruner's coupling constraints.

By explicitly estimating importance per language and aggregating with a Max policy, Lang-Prune avoids pruning decisions that are optimal on average yet harmful to minority or script-diverse languages. The framework is compute-efficient—importance estimation scales linearly with $|\mathcal{L}|$ using small calibration subsets—and is a drop-in multilingual extension of LLM-Pruner: sparsity schedules and prunable units remain unchanged, only the scoring and aggregation are revised.

## 3 Experiments

**Experiment Settings**  We evaluate Lang-Prune on `aya-expanse-8b` Dang et al. (2024) at 30%, 50%, and 70% global sparsity and compare it against LLM-Pruner under both monolingual and multilingual calibration. For coupled structure discovery and language-aware importance estimation, we construct a multilingual subset of mC4 Xue et al. (2021) and, for each language, sample 100 sequences of length 128: Arabic, Czech, German, English, Spanish, Indonesian, Hebrew, Russian, and Chinese.[1] Multilingual calibration uses a uniform per-language mixture (900 sequences total). Pruning is one-shot with no recovery. Following Section 2, Lang-Prune uses per-language min–max normalization and Max aggregator by default; Mean aggregator and Min aggregator are included as ablations. We evaluate PPL on the mC4 validation split with the same tokenizer and context length as calibration. All runs use a single NVIDIA A800 GPU, with each pruning instance requiring less than one GPU hour.

### 3.1 Results Analysis

Table 2 reports per-language PPL across three sparsity levels for *aya-expanse-8b* [2]. We analyze the results along three key dimensions:

**1. Overall performance improvement.** Lang-Prune-Max consistently outperforms LLM-Pruner under multilingual calibration for every language and sparsity. Relative to LLM-Pruner mixed-data, Lang-Prune-Max reduces average PPL by 14.6% (30%), 41.0% (50%), and 62.4% (70%). Compared to LLM-Pruner monolingual, Lang-Prune-Max achieves lower average PPL at all sparsities (14.90 vs 16.04 at 30%; 25.91 vs 29.16 at 50%; 70.85 vs 83.08 at 70%). On a per-language basis, Lang-Prune-Max surpasses the monolingual baseline in 5/9 languages at each sparsity (notably de, cs, en, es, id) while remaining competitive on the others. This improvement arises from per-language importance estimation and Max aggregation, which preserve structures critical to any language while pruning universally unimportant neurons.

**2. Worst-case language improvement.** Lang-Prune-Max also improves the worst-case performance across languages. At 70% sparsity, the worst-language PPL decreases from 378.39 (LLM-Pruner mixed-data, en) and 236.44 (LLM-Pruner monolingual, en) to 158.33 (Lang-Prune-Max, en). Similar trends hold at 30% and 50% sparsity. By explicitly protecting per-language critical structures, Lang-Prune reduces the risk that low-resource languages are disproportionately harmed during pruning.

---

[1] Abbreviations used throughout: ar, cs, de, en, es, id, iw, ru, zh.

[2] Detailed results for monolingual pruning can be found in Table 6 in Appendix A.1.

Table 2: Per-language perplexity (PPL, lower is better) after structured pruning of *aya-expanse-8b*. Lang-Prune variants merge per-language importance via Max (**Lang-Prune-Max**), Mean (**Lang-Prune-Avg**), or Min (**Lang-Prune-Min**) aggregator. For comparison, LLM-Pruner is evaluated under monolingual and multilingual (mixed-data) calibration.

| Method | Calibration | ar | cs | de | en | es | id | iw | ru | zh | Avg. ↓ |
|---|---|---|---|---|---|---|---|---|---|---|---|
| Original (8B) | None | 8.10 | 10.46 | 10.66 | 10.50 | 9.58 | 12.77 | 11.19 | 11.13 | 10.27 | **10.52** |
| *aya-expanse-8b with 30%* | | | | | | | | | | | |
| LLM-Pruner | monolingual | 11.10 | 15.54 | 15.10 | 30.50 | 16.38 | 15.60 | 13.58 | 13.76 | 12.84 | 16.04 |
| | mixed-data | 11.57 | 16.93 | 16.11 | 30.36 | 18.51 | 17.54 | 14.28 | 14.69 | 16.97 | 17.44 |
| Lang-Prune-Avg | multilingual | 11.47 | 15.80 | 14.76 | 24.78 | 16.74 | 16.25 | 14.25 | 14.13 | 15.71 | 15.99 |
| Lang-Prune-Min | multilingual | 354.12 | 612.46 | 284.30 | 66.85 | 182.67 | 262.45 | 1925.36 | 675.20 | 562.30 | 547.30 |
| **Lang-Prune-Max** | multilingual | **11.22** | **15.15** | **13.68** | **21.52** | **15.73** | **15.35** | **13.98** | **14.09** | **13.35** | **14.90** |
| *aya-expanse-8b with 50%* | | | | | | | | | | | |
| LLM-Pruner | monolingual | 17.48 | 28.34 | 28.39 | 70.76 | 29.64 | 26.71 | 21.10 | 22.09 | 17.93 | 29.16 |
| | mixed-data | 21.56 | 37.84 | 43.30 | 103.49 | 49.98 | 40.83 | 24.73 | 29.55 | 43.85 | 43.90 |
| Lang-Prune-Avg | multilingual | 20.73 | 31.11 | 32.68 | 71.03 | 35.65 | 31.97 | 24.34 | 24.74 | 38.61 | 34.54 |
| Lang-Prune-Min | multilingual | 757.21 | 1606.93 | 864.86 | 229.03 | 666.20 | 1053.59 | 26134.73 | 2618.83 | 1924.89 | 3984.03 |
| **Lang-Prune-Max** | multilingual | **17.99** | **25.63** | **23.02** | **49.20** | **26.52** | **25.58** | **22.22** | **22.78** | **20.30** | **25.91** |
| *aya-expanse-8b with 70%* | | | | | | | | | | | |
| LLM-Pruner | monolingual | 42.71 | 81.99 | 82.82 | 236.44 | 85.19 | 77.52 | 46.87 | 57.73 | 36.47 | 83.08 |
| | mixed-data | 80.99 | 164.12 | 245.65 | 378.39 | 227.26 | 173.63 | 71.97 | 127.51 | 226.88 | 188.49 |
| Lang-Prune-Avg | multilingual | 78.57 | 126.19 | 185.68 | 292.57 | 179.81 | 144.15 | 66.51 | 95.47 | 251.11 | 157.78 |
| Lang-Prune-Min | multilingual | 1588.60 | 2890.65 | 1973.42 | 579.72 | 1334.47 | 5097.47 | 56508.07 | 6862.74 | 3904.55 | 8960.00 |
| **Lang-Prune-Max** | multilingual | **44.03** | **69.71** | **64.85** | **158.33** | **72.53** | **70.08** | **51.62** | **60.29** | **46.18** | **70.85** |

**3. Insights from Lang-Prune-Min (negative control).** The Min aggregator, which takes the minimum importance across languages, serves as a negative-control ablation to examine the behavior of language-agnostic neurons. With Min aggregator, most languages experience severe performance degradation, while English—the dominant language—remains relatively robust. This indicates that even neurons considered language-agnostic carry residual bias toward dominant languages, and highlights the necessity of a language-aware aggregation strategy (e.g., Max aggregator) to maintain balanced multilingual performance.

Overall, these analyses demonstrate that Lang-Prune not only improves average performance but also mitigates worst-case outcomes and explicitly addresses language bias, providing more balanced multilingual pruning.

## 3.2 ANALYSIS OF PRUNING NEURON GROUPS

To compare the inner mechanisms of LLM-Pruner and Lang-Prune, we analyze the retention of language-specific capacity at high sparsity. We first identify, on the unpruned model, the set of strong language-related neuron groups per language and then measure their recall under different pruning strategies. Concretely, let $\{\mathcal{C}_j\}$ denote the coupled MLP channels, where $j$ uniquely identifies each candidate structure, and $\tilde{I}_\ell(\mathcal{C}_j)$ the per-language, min–max normalized importance from Section 2. For each language $\ell$, we define a contrastive specificity score for structure $\mathcal{C}_j$:

$$S_\ell(\mathcal{C}_j) = \tilde{I}_\ell(\mathcal{C}_j) - \frac{1}{|\mathcal{L}| - 1} \sum_{\ell' \neq \ell} \tilde{I}_{\ell'}(\mathcal{C}_j).$$

We construct the set of strong language-related groups by selecting the top-$p\%$ structures per language according to $S_\ell$ (with a fixed $p$ across languages). The recall ratio for language $\ell$ under a pruning strategy is defined as

$$\text{Recall}_\ell = \frac{\left|\{\mathcal{C}_j \in \text{Top-}p\% \text{ for } \ell\} \cap \{\mathcal{C}_j \text{ retained after pruning}\}\right|}{\left|\{\mathcal{C}_j \in \text{Top-}p\% \text{ for } \ell\}\right|}.$$

This metric quantifies how well a pruning method preserves neuron groups most specialized to each language, independent of the method's own scoring.

Table 3 reports the recall ratio at 70% sparsity with Top-$p\%$ = 30% across nine languages. Compared with multilingual mixed-data pruning (LLM-Pruner mixed-data), Lang-Prune consistently retains a

Table 3: Recall ratio of strong language-related neuron groups (Top-$p\%$ = 30%) under different pruning strategies at 70% sparsity. Higher is better.

| Method | ru | iw | id | es | en | de | cs | ar | zh | Avg. |
|---|---|---|---|---|---|---|---|---|---|---|
| Mixed-data | 67.47% | 69.20% | 67.43% | 65.06% | 60.06% | 66.22% | 69.68% | 66.79% | 61.74% | 65.96% |
| Lang-Prune (ours) | 81.94% | 82.86% | 81.62% | 80.68% | 75.41% | 81.71% | 83.11% | 81.27% | 80.78% | 81.04% |

larger fraction of language-specific neurons, improving the average recall from 65.96% to 81.04%. Gains are broad (e.g., zh: 61.74% → 80.78%; cs: 69.68% → 83.11%), indicating that Max aggregation protects structures that are critical to any language rather than optimizing for average-case activation. These retention improvements align with the perplexity results in Table 6, and are most pronounced in languages that showed higher interference under mixed calibration.

## 4 ABLATION STUDIES

### 4.1 GENERALIZATION ACROSS MODEL TYPES

To assess model generality, we apply Lang-Prune to an additional multilingual LLM, `Qwen3-8B` Yang et al. (2025), which differs in tokenizer and architectural choices from `aya-expanse-8b`. For each model, we replicate the setup from Section 3: sparsities at 30%, 50%, and 70%; identical calibration/evaluation protocol and comparisons against LLM-Pruner under monolingual and multilingual calibration.

Table 4: Cross-lingual Interference Factor (IF) on `Qwen3-8B` across sparsities. $IF(l) = PPL_{Multi}(l)/PPL_{Mono}(l)$, computed relative to the LLM-Pruner monolingual baseline (lower is better; IF $< 1$ indicates improvement). Rows report per-language PPL; IF rows report the ratio vs Monolingual.

| PPL ↓ | ar | cs | de | en | es | id | iw | ru | zh | Avg. |
|---|---|---|---|---|---|---|---|---|---|---|
| *Qwen3-8B at 30% sparsity* | | | | | | | | | | |
| LLM-Pruner (monolingual) | 13.53 | 9.05 | 13.62 | 31.19 | 13.93 | 9.41 | 24.22 | 8.50 | 14.37 | 15.76 |
| LLM-Pruner (mixed-data) | 15.76 | 9.82 | 13.05 | 24.31 | 14.87 | 9.99 | 31.06 | 8.93 | 16.18 | 16.00 |
| *IF vs monolingual* | 1.16× | 1.09× | 0.96× | 0.78× | 1.07× | 1.06× | 1.28× | 1.05× | 1.13× | 1.06× |
| Lang-Prune | 14.19 | 9.24 | 12.00 | 21.24 | 13.39 | 9.20 | 26.23 | 8.46 | 13.13 | 14.12 |
| *IF vs monolingual* | 1.05× | 1.02× | 0.88× | 0.68× | 0.96× | 0.98× | 1.08× | 0.99× | 0.91× | 0.90× |
| *Qwen3-8B at 50% sparsity* | | | | | | | | | | |
| LLM-Pruner (monolingual) | 20.28 | 14.77 | 35.92 | 118.76 | 29.96 | 20.69 | 37.05 | 17.93 | 35.11 | 36.94 |
| LLM-Pruner (mixed-data) | 208.99 | 102.79 | 207.90 | 565.88 | 205.65 | 144.37 | 723.06 | 61.84 | 575.77 | 310.69 |
| *IF vs monolingual* | 10.31× | 6.96× | 5.79× | 4.77× | 6.86× | 6.98× | 19.52× | 3.45× | 16.40× | 9.00× |
| Lang-Prune | 22.87 | 14.57 | 19.62 | 44.50 | 22.02 | 14.65 | 44.87 | 13.25 | 22.35 | 24.30 |
| *IF vs monolingual* | 1.13× | 0.99× | 0.55× | 0.38× | 0.73× | 0.71× | 1.21× | 0.74× | 0.64× | 0.66× |
| *Qwen3-8B at 70% sparsity* | | | | | | | | | | |
| LLM-Pruner (monolingual) | 73.10 | 134.00 | 752.05 | 948.83 | 391.40 | 344.82 | 117.63 | 179.41 | 559.01 | 388.69 |
| LLM-Pruner (mixed-data) | 99441.44 | 35822.38 | 46402.90 | 15611.48 | 35874.89 | 28035.05 | 180241.13 | 62441.93 | 28241.14 | 53567.15 |
| *IF vs monolingual* | 1360.35× | 267.33× | 61.70× | 16.45× | 91.66× | 81.30× | 1532.27× | 348.04× | 50.52× | 137.85× |
| Lang-Prune | 126.94 | 57.09 | 122.29 | 400.39 | 151.12 | 86.70 | 339.80 | 56.10 | 230.26 | 174.74 |
| *IF vs monolingual* | 1.74× | 0.43× | 0.16× | 0.42× | 0.39× | 0.25× | 2.89× | 0.31× | 0.41× | 0.45× |

**Observations.** (i) At 30% sparsity, Lang-Prune improves average IF to 0.90 and reduces PPL in most languages versus the monolingual baseline, while LLM-Pruner (mixed-data) slightly degrades (avg IF 1.06). (ii) At 50%, LLM-Pruner (mixed-data) suffers severe cross-lingual interference (avg IF 9.00), whereas Lang-Prune maintains IF well below 1 on average (0.66), indicating robustness under tighter budgets. (iii) At 70%, LLM-Pruner (mixed-data) exhibits pathological degradation on `Qwen3-8B` (extreme PPL), suggesting instability of mixed-language scoring with strongly coupled structures on this model. By contrast, Lang-Prune remains stable and substantially below the monolingual baseline on average (avg IF 0.45), though a few languages (e.g., iw) still show $IF > 1$.

**Takeaway.** These results demonstrate that Lang-Prune generalizes effectively across model families with different tokenization and architectural designs. While LLM-Pruner under multilingual calibration becomes increasingly unstable—especially at higher sparsity—Lang-Prune consistently suppresses cross-lingual interference and maintains robust performance, validating its portability and resilience beyond a single model backbone. We further evaluated Lang-Prune across a broader

range of model scales, confirming that its benefits persist from mid-size to large models; detailed results are provided in Appendix A.8.

## 4.2 POST-TRAINING POTENTIAL AFTER PRUNING

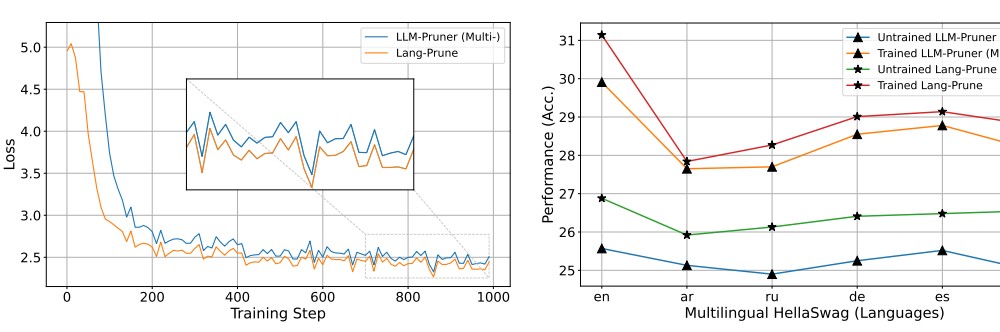

(a) Training Loss Curves on pruned Qwen3-8B.

(b) Few-shot Evaluation for Post-training

Figure 2: Post-training recovery of pruned models: (a) Training loss during continued pre-training of Qwen3-8B at 70% sparsity. Lang-Prune consistently achieves lower loss than LLM-Pruner (mixed-data) across steps; (b) Few-shot accuracy before and after LoRA post-training for models pruned by LLM-Pruner (mixed-data) and Lang-Prune. Lang-Prune yields higher pre- and post-training accuracy and larger recovery across languages.

We evaluate whether Lang-Prune preserves the capacity of pruned models to benefit from short post-training. Starting from 70% sparse `Qwen3-8B` pruned checkpoints, we apply an identical fine-tuning budget to each method and measure: (1) **language-modeling recovery**, using $\Delta\text{PPL} = \text{PPL}_{\text{pre}} - \text{PPL}_{\text{post}}$, as shown in Figure 2a; and (2) **downstream task recovery**, using $\Delta\text{Acc} = \text{Acc}_{\text{post}} - \text{Acc}_{\text{pre}}$ on the 3-shot multilingual benchmark translated-HellaSwag Dac Lai et al. (2023); Zellers et al. (2019) (Figure 2b). Detailed few-shot results are provided in Table 7 in Appendix A.1.

**Continued pre-training setup (shared across methods): Parameter-efficient:** LoRA with rank $r = 16$ and $\alpha = 32$, applied to all MLP and attention modules, with base weights frozen. **Training protocol:** Models are trained on Wikipedia Foundation with a sequence length of 256; an effective batch size of 1M tokens per step is achieved via gradient accumulation. Training is run for 1,000 steps (1B tokens in total) using the HuggingFace Trainer with identical default optimization settings.

**Observations.** (i) Lang-Prune consistently achieves higher few-shot accuracy than LLM-Pruner across all reported languages, both before and after continuous post-training, indicating better preservation of trainable capacity after pruning (see Figure 2b). (ii) With LoRA post-training, Lang-Prune further improves over LLM-Pruner, consistent with the lower training loss in Figure 2a. Protecting language-critical structure enables more efficient adaptation under parameter-efficient tuning. (iii) Gains are largest in languages that previously exhibited higher cross-lingual interference (e.g., en, ru, id), aligning with Lang-Prune's objective of mitigating interference while retaining recoverable capacity.

**Takeaway.** Lang-Prune not only reduces cross-lingual interference during pruning but also preserves the model's post-training potential. Even under a limited fine-tuning budget, Lang-Prune checkpoints recover more quickly and achieve higher downstream performance than LLM-Pruner, demonstrating that language-aware structure preservation yields pruned models that remain adaptable and robust across languages. The additional multilingual benchmarks reported in Appendix A.6 further confirm these conclusions.

## 4.3 GENERALIZATION TO OUT-OF-DISTRIBUTION LANGUAGES

We evaluate zero-shot generalization to languages absent from calibration (OOD). Using the 70% structured pruning setting, we compare: (i) LLM-Pruner with mixed-language calibration (mixed-data), (ii) the best monolingual proxy among the nine in-distribution languages (Best-

mono), and (iii) Lang-Prune. We report per-language PPL and the generalization ratio $GR(l) = PPL_{Lang-Prune}(l)/PPL_{Data-mixed}(l)$, where values below 1 favor Lang-Prune.

Table 5: OOD languages at 70% sparsity. Family/Script tags are included for interpretability.

| Language | Family / Script | Best mono source | PPL (Mixed-data) | PPL (Best-mono) | PPL (Lang-Prune) | GR ↓ |
|---|---|---|---|---|---|---|
| fa (Persian) | Indo-Iranian / Arabic | ar | 154.37 | 92.90 | 62.26 | 0.40 |
| ur (Urdu) | Indo-Aryan / Arabic | iw | 273.27 | 107.91 | 31.08 | 0.11 |
| am (Amharic) | Semitic / Ethiopic | zh | 258.83 | 12.25 | 11.90 | 0.05 |
| bg (Bulgarian) | Slavic / Cyrillic | ru | 128.64 | 91.21 | 71.05 | 0.55 |
| uk (Ukrainian) | Slavic / Cyrillic | ru | 152.68 | 67.09 | 57.91 | 0.38 |
| pl (Polish) | Slavic / Latin | cs | 199.85 | 104.12 | 68.24 | 0.34 |
| nl (Dutch) | Germanic / Latin | en | 410.68 | 178.78 | 104.86 | 0.26 |
| sv (Swedish) | Germanic / Latin | en | 404.02 | 291.75 | 144.52 | 0.36 |
| da (Danish) | Germanic / Latin | en | 440.81 | 280.81 | 160.90 | 0.37 |
| fr (French) | Romance / Latin | id | 318.40 | 143.33 | 78.00 | 0.25 |
| it (Italian) | Romance / Latin | id | 268.33 | 148.46 | 78.04 | 0.29 |
| pt (Portuguese) | Romance / Latin | id | 333.11 | 127.94 | 81.01 | 0.24 |
| my (Malay) | Austronesian / Latin | zh | 14.46 | 8.06 | 7.64 | 0.53 |
| ja (Japanese) | Japonic / Jpn (CJK) | zh | 608.95 | 158.51 | 155.66 | 0.26 |
| ko (Korean) | Koreanic / Hangul | zh | 346.27 | 89.62 | 76.57 | 0.22 |
| vi (Vietnamese) | Austroasiatic / Latin | zh | 303.86 | 149.50 | 76.76 | 0.25 |

**Observations.** **(i) Strong OOD gains:** Lang-Prune substantially outperforms the mixed-data baseline for every OOD language (GR average 0.30; $\sim 70\%$ mean reduction in PPL). The largest improvements occur for ur (0.11×), am (0.05×), ko (0.22×), and vi (0.25×). **(ii) Outperforming best monolingual proxies:** On average, Lang-Prune reduces PPL by $\sim 32\%$ compared to the best-of-mono proxy per OOD language (mean ratio $\sim 0.68$), indicating that preserving structures important to any in-distribution language transfers better than committing to a single source. **(iii) Family/script affinity patterns:** (1) Slavic/Cyrillic OOD languages (bg, uk) are best served by ru, and West Slavic (pl) by cs, matching family and script. (2) Germanic/Latin OOD (nl, sv, da) favor en, consistent with lexical and tokenization overlap in Latin scripts. (3) CJK/East Asian (ja, ko) and several SE-Asian cases (vi, my) favor zh; for ja, this is plausibly aided by shared Kanji; for vi/my, the effect likely stems from tokenization and segmentation biases rather than genealogical relatedness. (4) Arabic-script OOD (fa, ur) align with ar/iw proxies, reflecting script directionality and character set effects. (5) Romance/Latin OOD (fr, it, pt) favor id rather than es; this suggests that script-level overlap and morphological simplicity (shorter subwords, reduced inflection) can dominate genealogical proximity under pruning.

**Takeaway.** Lang-Prune's language-aware importance scoring provides robust OOD generalization, outperforming both mixed-language pruning and the best single-language proxy. Proxy selection correlates more with script and tokenization overlap than strict language family, highlighting that preserving diverse language-specific structures benefits transfer to unseen languages.

## 4.4 Conclusion on Ablation Studies

Across all ablations, Lang-Prune demonstrates robust and consistent improvements beyond standard perplexity comparisons. First, in model-type generalization (Section 4.1), the method transfers effectively to architectures and tokenizers distinct from `aya-expanse-8b` (e.g., `Qwen3-8B`), consistently reducing both average and worst-case PPL at 30%, 50%, and 70% sparsity while keeping IF well below 1, whereas mixed-language LLM-Pruner exhibits severe degradation at higher sparsity. Second, under identical post-training budgets with LoRA (Section 4.2), Lang-Prune preserves greater headroom for adaptation, yielding higher few-shot accuracies across languages and more efficient recovery per token, supporting the hypothesis that protecting language-critical structures facilitates downstream tuning. Third, in zero-shot transfer to out-of-distribution languages (Section 4.3, Table 5), Lang-Prune substantially outperforms mixed-language pruning (mean $GR \approx 0.30$) and even surpasses the best monolingual proxy on average. Proxy analysis indicates that transfer is driven more by script and tokenization overlap than by strict genealogical relatedness, highlighting the value of preserving diverse, language-specific structures during pruning.

Besides the ablation studies discussed above, we also observe the following: (i) Applying Lang-Prune in multi-task or multi-domain settings yields only marginal improvements, which we attribute to the much weaker structural separation between tasks compared to languages (see Appendix A.4). (ii) Lang-Prune achieves consistent gains across different calibration dataset sizes. Specifically, increasing the monolingual calibration set from 100 to 900 sequences improves performance but still falls short of Lang-Prune (see Appendix A.5).

## 5 RELATED WORKS

### 5.1 LLM PRUNING

Pruning reduces inference cost by removing parameters while preserving accuracy. Unstructured, post-training methods such as Wanda Sun et al. (2024) and SparseGPT Frantar & Alistarh (2023) achieve high sparsity with minimal or no retraining via activation-aware magnitude or second-order criteria. Structured pruning removes entire components (e.g., MLP channels, attention heads) for deployment-friendly speedups Michel et al. (2019); Lagunas et al. (2021); Fan et al. (2019); Sajjad et al. (2023). LLM-Pruner Ma et al. (2023) formalizes structured pruning using dependency graphs and coupled structures. Most prior methods are single-dataset and language-agnostic; Lang-Prune extends LLM-Pruner by estimating importance per language and aggregating via a multilingual Max rule to protect critical structures.

### 5.2 MULTILINGUAL PRUNING AND LANGUAGE-AWARE COMPRESSION

Multilingual pruning shows heterogeneous effects across languages and tasks Ogueji et al. (2022). Calibration with multiple languages can help at moderate sparsity, though results vary Zeng et al. (2024); Kurz et al. (2024). Notably, Multilingual Brain Surgeon (MBS) Zeng et al. (2024) samples calibration data in language-balanced mixtures while keeping a single shared pruning criterion, which partially mitigates language bias but does not change how importance is computed. Alignment-informed methods such as Kim et al. (2024) leverage bilingual or translation-style signals to guide pruning, targeting multilingual inference with supervision. Lang-Prune differs by preserving per-language importance and aggregating via a worst-case Max rule, directly reducing cross-lingual interference.

### 5.3 LANGUAGE-SPECIFIC AND UNIVERSAL STRUCTURE IN MULTILINGUAL MODELS

Multilingual transformers combine shared and language-specific mechanisms: lower/middle layers encode form and morpho-syntax, while higher layers are more semantic and language-agnostic Belinkov & Glass (2019); Tenney et al. (2019); Pires et al. (2019); Conneau et al. (2020). Certain neurons or subspaces align with scripts or linguistic features Singh et al. (2019); Liu et al. (2019). Preserving language-specific subnetworks supports cross-lingual transfer Choenni et al. (2022). Lang-Prune builds on this by protecting units critical to any language while pruning universally unimportant structures, outperforming uniform or MBS-style calibration sampling in one-shot structured pruning (see Appendix A.3).

## 6 CONCLUSION

In this paper, we investigate cross-lingual interference in pruning multilingual LLMs and propose Lang-Prune, a drop-in extension to LLM-Pruner that computes per-language importance (min–max normalized) and aggregates with a Max rule to protect units critical to any language. The pilot quantifies interference under mixed-language calibration; Lang-Prune mitigates it and consistently improves average and worst-language performance across sparsities on aya-expanse-8b (e.g., at 70% sparsity: average PPL 70.85 vs 188.49 for multilingual LLM-Pruner, and 70.85 vs 83.08 for monolingual). Interpretability analyses show higher retention of language-specific capacity (recall 81.0% vs 66.0%). Ablations indicate robustness across model types (e.g., `Qwen3-8B`), better post-training headroom (including with LoRA), and strong transfer to OOD languages (average $GR \approx 0.30$). Lang-Prune is compute-efficient and deployment-friendly.

## LIMITATIONS

While Lang-Prune demonstrates strong multilingual performance, several limitations warrant discussion. First, our approach employs only basic aggregation strategies (max/min/mean); more sophisticated methods such as weighted combination based on language characteristics remain unexplored. Second, the framework is evaluated primarily in one-shot pruning settings without extensive recovery, leaving combinations with quantization, distillation, or prolonged fine-tuning for future work. Third, the method requires full model access for activation collection, limiting its applicability to proprietary or black-box LLMs. Furthermore, Lang-Prune's effectiveness depends on structured pruning paradigms. As shown in Appendix A.7, the method does not improve performance with unstructured pruning approaches like Wanda, suggesting it relies on semantically meaningful structural units rather than individual weights.

Future work should explore adaptive aggregation strategies, integration with diverse compression techniques, and broader evaluation across languages, architectures, and pruning granularities.

## ETHICS STATEMENT

This study relies exclusively on fully open-source text datasets. All datasets had been comprehensively anonymized by their original providers prior to our use, ensuring the absence of any personally identifiable information. Consequently, the utilization of these datasets does not involve any infringement of individual privacy. The LLMs pruning framework introduced in this paper is designed strictly for academic and scientific research purposes. Any application of this framework must adhere to established legal regulations and ethical standards. The authors explicitly prohibit its deployment in unlawful activities or in any manner that could cause harm to individuals or society.

## REPRODUCIBILITY STATEMENT

This study is committed to ensuring the reproducibility of its findings. To guarantee full transparency of the data, methods, and experimental procedures, all experiments are conducted using publicly accessible datasets, as detailed in Section 2, Section 3 and Section 4. Comprehensive descriptions of the framework design, performance evaluation, and experimental setup are provided in Section 2, Section 3 and Section 4. Furthermore, the complete codebase (including training and inference configurations) will be released on GitHub upon the full acceptance of this paper, enabling the research community to replicate our results.

## LARGE LANGUAGE MODELS USAGE STATEMENT

For this work, we used large language models (LLMs) as a general-purpose assistive tool to improve clarity, grammar, and phrasing in portions of the manuscript. Specifically, ChatGPT was employed to: (1) Suggest alternative phrasings for sentences and paragraphs to enhance readability. No part of the research ideas, results, or technical contributions was generated by LLMs. All scientific content, including experiments, analysis, and conclusions, was independently conceived and verified by the authors.

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

# A  APPENDIX

## A.1  EXPERIMENT RESULT DETAILS

Table 6: Per-language perplexity after 70% structured pruning. Lang-Prune merges per-language importance via Max (default), compared to mean and min. We also report LLM-Pruner under monolingual and multilingual (mixed-data) calibration.

| Method | Calibration | ar | cs | de | en | es | id | iw | ru | zh | Avg. ↓ |
|---|---|---|---|---|---|---|---|---|---|---|---|
| Original (8B) | None | 8.10 | 10.46 | 10.66 | 10.50 | 9.58 | 12.77 | 11.19 | 11.13 | 10.27 | **10.52** |
| LLM-Pruner (70%) | ar | **42.71** | 1933.60 | 1045.78 | 602.23 | 651.80 | 1361.45 | 297.98 | 1038.27 | 1121.41 | 899.47 |
| | cs | 347.90 | **81.99** | 236.12 | 339.14 | 244.47 | 272.87 | 883.65 | 130.36 | 998.25 | 392.75 |
| | de | 497.99 | 503.61 | **82.82** | 365.89 | 309.53 | 342.55 | 2629.72 | 697.91 | 1426.27 | 761.81 |
| | en | 1166.51 | 2090.46 | 1165.79 | **236.44** | 705.24 | 2062.33 | 64235.58 | 4802.09 | 3246.88 | 8856.81 |
| | es | 404.32 | 532.11 | 422.02 | 340.05 | **85.19** | 354.68 | 1764.65 | 631.21 | 1376.49 | 656.75 |
| | id | 333.15 | 482.84 | 328.63 | 293.57 | 252.14 | **77.52** | 1418.63 | 594.34 | 713.24 | 499.34 |
| | iw | 84.05 | 628.21 | 481.31 | 360.53 | 326.31 | 414.81 | **46.87** | 252.18 | 372.41 | 329.63 |
| | ru | 252.89 | 165.36 | 337.70 | 336.58 | 261.16 | 323.61 | 506.07 | **57.73** | 450.77 | 299.10 |
| | zh | 271.20 | 878.06 | 591.66 | 368.54 | 495.26 | 371.47 | 629.59 | 705.54 | **36.47** | 483.09 |
| | monolingual | 42.71 | 81.99 | 82.82 | 236.44 | 85.19 | 77.52 | 46.87 | 57.73 | 36.47 | 83.08 |
| | mixed-data | 80.99 | 164.12 | 245.65 | 378.39 | 227.26 | 173.63 | 71.97 | 127.51 | 226.88 | 188.49 |
| Lang-Prune-Avg (70%) | multilingual | 78.57 | 126.19 | 185.68 | 292.57 | 179.81 | 144.15 | 66.51 | 95.47 | 251.11 | 157.78 |
| Lang-Prune-Min (70%) | multilingual | 1588.60 | 2890.65 | 1973.42 | 579.72 | 1334.47 | 5097.47 | 56508.07 | 6862.74 | 3904.55 | 8960.00 |
| **Lang-Prune-Max (70%)** | multilingual | **44.03** | **69.71** | **64.85** | **158.33** | **72.53** | **70.08** | **51.62** | **60.29** | **46.18** | **70.85** |

Table 7: 3-shot Multilingual translated-HellaSwag accuracy ↑ (%) on `Qwen3-8B` after identical fine-tuning budgets. Mean±std over seeds. Lang-Prune preserves more post-training headroom than LLM-Pruner, with consistent gains when using LoRA.

| Method | en | ar | ru | de | es | id |
|---|---|---|---|---|---|---|
| **Original** | 57.21±0.49 | 39.07±0.51 | 44.91±0.52 | 45.74±0.51 | 49.65±0.52 | 44.85±0.52 |
| **LLM-Pruner (mixed-data)** | | | | | | |
| *- w/o training* | 25.57±0.44 | 25.13±0.45 | 24.90±0.45 | 25.25±0.45 | 25.52±0.45 | 25.02±0.45 |
| *- w LoRA* | 29.91±0.46 | 27.65±0.47 | 27.70±0.46 | 28.55±0.47 | 28.78±0.47 | 28.18±0.47 |
| **Lang-Prune** | | | | | | |
| *- w/o training* | 26.88±0.44 | 25.92±0.46 | 26.13±0.46 | 26.41±0.46 | 26.48±0.46 | 26.55±0.46 |
| *- w LoRA* | **31.14±0.46** | **27.84±0.47** | **28.27±0.47** | **29.01±0.47** | **29.14±0.47** | **28.81±0.47** |

## A.2  ABLATION STUDIES ON GROUP AGGREGATOR

We investigate how the choice of group aggregator $\mathcal{A}$ affects pruning behavior. Recall that $\mathcal{A}$ reduces parameter- or tensor-level importance scores within each coupled structure. In addition to our default `sum` aggregator, we evaluate $\mathcal{A} = \{\text{mean}, \text{max}, \text{first}\}$.

- **sum**: accumulates the total contribution of all parameters.
- **mean**: normalizes by group size to reduce the effect of large structures.
- **max**: highlights structures where a single parameter dominates.
- **first**: uses the first entry as a simplified representative value.

Table 8 reports average perplexity across pruning strategies (Monolingual, Mixed-data, and Lang-Prune) using `Qwen3-8B` with 50% sparsity. All other settings follow Section 3.

Table 8: Ablation on group aggregators. Lower is better.

| Avg PPL ↓ | sum | mean | max | first |
|---|---|---|---|---|
| Monolingual | 36.72 | 33.28 | 20.74 | 381.29 |
| Mixed-data | 310.69 | 126.83 | 29.29 | 6122.22 |
| Lang-Prune (ours) | **24.30** | **24.24** | **22.53** | **45.23** |

These results show that although alternative aggregators may work reasonably in certain settings, `sum` and `mean` provide the most stable performance across multilingual scenarios. `max` offers

more aggressive sparsity behavior, while `first` produces unstable behavior and is therefore not recommended.

## A.3 MBS-STYLE CALIBRATION COMPARISON

To evaluate the impact of multilingual calibration strategies, we approximate the Multilingual Brain Surgeon (MBS) Zeng et al. (2024) by sampling calibration data with various language mixture ratios; exact recipes are not available for the Qwen and Aya models. We compare these with uniform mixtures and Lang-Prune's language-aware Max aggregation on `Qwen3-8B` at 50% sparsity.

Table 9: Per-language perplexity (PPL)↓ for different calibration strategies. Bold indicates best. Underline indicates best among MBS-style mixtures.

| Avg PPL ↓ | EN:Others 3:1 | EN:Others 2:1 | EN:Others 5:1 | EN:ZH:ES:Others 3:2:2:1 | EN:ZH:ES:Others 4:2:2:1 | Uniform | Lang-Prune |
|---|---|---|---|---|---|---|---|
| ar | 506.51 | 110.02 | 298.05 | 313.12 | 331.37 | 208.99 | **22.87** |
| cs | 211.97 | 80.93 | 386.84 | 182.11 | 159.93 | 102.79 | **14.57** |
| de | 515.74 | 157.60 | 644.36 | 405.21 | 291.29 | 207.90 | **19.62** |
| en | 1055.65 | 364.47 | 829.81 | 976.80 | 753.70 | 565.88 | **44.50** |
| es | 379.91 | 143.36 | 411.79 | 326.87 | 289.16 | 205.65 | **22.02** |
| id | 366.43 | 100.47 | 229.31 | 270.06 | 230.32 | 144.37 | **14.65** |
| iw | 1226.97 | 539.44 | 3167.02 | 1791.35 | 1333.17 | 723.06 | **44.87** |
| ru | 179.46 | 51.07 | 168.75 | 152.53 | 106.46 | 61.84 | **13.25** |
| zh | 722.71 | 617.56 | 1318.28 | 629.44 | 528.75 | 575.77 | **22.35** |
| avg | 573.93 | 240.55 | 828.24 | 560.83 | 447.13 | 310.69 | **24.30** |

As results shown in Table 9, MBS-style rebalancing helps versus uniform mixtures, but Lang-Prune's per-language scoring with Max aggregation yields substantially better perplexities (avg 24.30 vs best MBS 240.55, ∼10× improvement).

## A.4 MULTILINGUAL VS. MULTI-DOMAIN PRUNING

While the high-level idea behind language-aware pruning resembles task- or domain-aware pruning, the underlying structure differs fundamentally. Human languages are highly distinct in vocabulary, syntax, morphology, and script, whereas closely related tasks often share significant latent structure, vocabulary, and reasoning patterns. Consequently, multilingual models naturally organize **discrete, language-specific neuron groups**, as observed in prior work Tan et al. (2024); Wang et al. (2025); Tan et al. (2024). These neuron clusters are highly separable, providing a clear signal for structure-preserving pruning strategies such as Lang-Prune. In contrast, multi-task or multi-domain settings typically involve subtle differences. Tasks or domains that share vocabulary and reasoning patterns produce overlapping activations, making per-task neuron importance less stable. This structural difference explains why techniques effective in multilingual pruning may not directly translate to task-aware pruning.

To empirically validate this, we applied a similar *per-domain importance + max aggregation* strategy to a multi-domain scenario using the `Qwen3-8B` model. We treat three MMLU subfields—Law, Medical, and Finance—as domains. Calibration is performed using the respective train/dev splits, and evaluation is conducted on the test split at 50% sparsity.

Table 10: Domain-aware pruning results on Qwen3-8B (50% sparsity). Unlike multilingual pruning, domain-aware pruning provides only marginal improvements.

| Avg Acc ↑ (stderr) | Law | Medical | Finance |
|---|---|---|---|
| Base Model | 73.1% ± 13.8% | 76.4% ± 8.3% | 78.9% ± 12.7% |
| Mixed-data | 37.6% ± 6.6% | 35.3% ± 6.7% | 37.5% ± 5.3% |
| Lang-Prune (ours) | 40.1% ± 9.4% | 37.7% ± 7.7% | 37.8% ± 7.1% |

As shown in Table 10, domain-aware pruning achieves only marginal improvements over mixed-data pruning and remains far below the base model. Unlike the multilingual scenario, the differences

between domains are not sufficiently large to yield stable per-domain importance rankings. This instability leads to limited or inconsistent benefits from max-aggregation strategies.

These observations highlight that **language-aware pruning is fundamentally different from task-aware pruning**: the strong, discrete separation of neuron groups in multilingual LLMs enables reliable, structured pruning strategies like Lang-Prune, which cannot be trivially generalized to multi-domain or multi-task scenarios.

## A.5 INFLUENCE OF CALIBRATION SIZE

Table 11: Comparison of multilingual and monolingual pruning with different calibration sizes. "N each" denotes number of sequences per language. Lang-Prune uses the same total calibration size as mixed-data (9N total).

| Method (num sample) | Sparsity | N=10 | N=30 | N=50 | N=80 | N=100 | N=150 | N=200 |
|---|---|---|---|---|---|---|---|---|
| Mixed-data (9N total) | 0.3 | 14.55 | 14.29 | 15.02 | 14.91 | 16.00 | 14.58 | 15.19 |
| Monolingual (N each) | 0.3 | 18.39 | 16.95 | 15.07 | 14.96 | 15.76 | 14.70 | 15.23 |
| Lang-Prune (ours, 9N total) | 0.3 | **13.57** | **13.53** | **13.39** | **13.35** | **14.12** | **13.44** | **13.44** |
| Mixed-data (9N total) | 0.5 | 34.83 | 35.70 | 44.72 | 76.68 | 310.69 | 331.62 | 384.20 |
| Monolingual (N each) | 0.5 | 77.40 | 54.17 | 36.70 | 34.12 | 36.94 | 32.70 | 36.73 |
| Lang-Prune (ours, 9N total) | 0.5 | **23.22** | **21.54** | **21.00** | **20.60** | **24.30** | **21.36** | **21.33** |
| Mixed-data (9N total) | 0.7 | 736.09 | 707.91 | 2406.78 | 19716.96 | 53567.15 | 16444.23 | 91970.55 |
| Monolingual (N each) | 0.7 | 1668.59 | 1009.15 | 432.54 | 348.63 | 388.69 | 254.56 | 247.01 |
| Lang-Prune (ours, 9N total) | 0.7 | **200.12** | **146.17** | **103.90** | **90.31** | **174.74** | **85.28** | **80.13** |

To evaluate the effect of calibration size, we ran additional monolingual pruning experiments using 900 calibration sequences per language, keeping all other settings identical. This allows comparison with Lang-Prune, which uses the same total calibration size across all languages. Table 11 reports the results for `Qwen3-8B` (base average PPL = 11.02) at 30%, 50%, and 70% sparsity.

The results show that monolingual pruning benefits from larger calibration sets but still consistently underperforms Lang-Prune, even when given the same per-language budget. At moderate sparsity (30–50%), increasing monolingual calibration size reduces perplexity, yet its performance plateaus quickly and remains notably worse than Lang-Prune across all settings. At high sparsity (70%), monolingual pruning becomes highly unstable: although more calibration data improves robustness, its perplexity remains 3–5× higher than Lang-Prune, which maintains strong performance even under extreme compression. In contrast, mixed-data pruning shows severe degradation—especially at higher sparsity—demonstrating that aggregating multilingual data without structure-aware separation produces highly unreliable pruning scores. These results confirm that Lang-Prune's improvement is primarily due to its design principle of protecting language-specific structures, rather than merely benefiting from a larger total calibration size.

### A.5.1 IMPORTANCE DISTRIBUTION UNDER VARYING CALIBRATION SIZE

As shown in Table 11, for both monolingual and mixed-data pruning, increasing the calibration size eventually leads to performance degradation, especially under high sparsity. This counter-intuitive trend raises an important question: *why does more calibration data harm pruning?*

To investigate this phenomenon, we analyze how the importance score distribution changes as the calibration size varies. Specifically, we compute normalized importance scores for coupled structures across all 36 layers of `Qwen3-8B`, using English-only calibration data and varying $N$ from 10 to 1800.

Figure 3 reveals a consistent pattern across layers: as $N$ increases, the importance scores become more uniformly distributed, with a clear increase in density in the mid-importance region (0.1–0.3). This reduces the contrast between critical and non-critical structures, making ranking-based pruning less discriminative. We refer to this phenomenon as **importance dilution**.

Under high sparsity, this dilution makes pruning unstable: if many components appear moderately important, aggressive pruning may remove genuinely critical structures while retaining mediocre

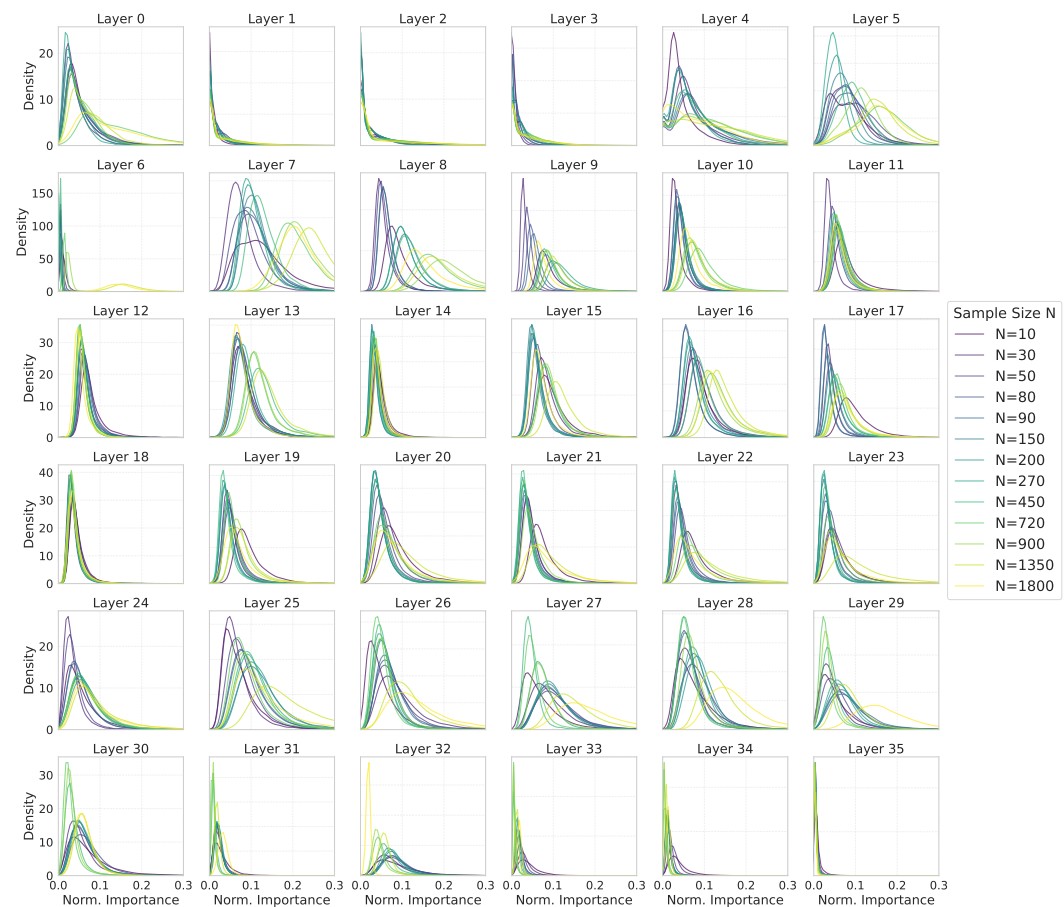

Figure 3: Impact of calibration size (N) on normalized importance score distribution across 36 layers (zoomed to 0–0.3). Larger N increases density in the mid-importance region, reducing score separability.

ones, resulting in sudden performance collapse. Mixed-data pruning is even more vulnerable due to higher topic and domain diversity, which further spreads residual importance.

In contrast, Lang-Prune remains robust because its Max aggregation selectively preserves peak language-specific signals, maintaining importance separability even when calibration size increases. This explains why larger calibration datasets do not yield further benefits for monolingual pruning and instead lead to degradation, while Lang-Prune remains stable.

### A.6 POST-TRAINING RECOVERY ON FEW-SHOT BENCHMARKS

We evaluate whether Lang-Prune preserves the ability of pruned models to benefit from brief post-training. Starting from pruned checkpoints, we apply an identical LoRA fine-tuning budget to each method and measure downstream task recovery. We follow the continued pre-training and LoRA setup described in Section 4.2 of the main text. All methods start from 70% pruned `Qwen3-8B` checkpoints and use identical LoRA fine-tuning budgets. All evaluations are performed under 3-shot settings.

In addition to *translated-HellaSwag* (commonsense reasoning) Dac Lai et al. (2023); Zellers et al. (2019), we evaluate on multilingual benchmarks covering comprehension, reasoning, and knowledge: *Belebele* Bandarkar et al. (2023) (multilingual reading comprehension), *translated-ARC* Dac Lai et al. (2023); Clark et al. (2018) (grade-school reasoning, multi-language), and *Global-MMLU* Singh et al. (2024) (broad multilingual knowledge). The languages used for each benchmark are listed in Table 12.

Table 12: Languages for Few-shot Benchmarks.

| Benchmark | Languages |
|---|---|
| Belebele | Arabic, Hebrew, Czech, Russian, German, Spanish, Indonesian, Chinese |
| translated-ARC | English, Arabic, Russian, German, Spanish, Indonesian, Chinese |
| translated-HellaSwag | English, Arabic, Russian, German, Spanish, Indonesian |
| Global-MMLU | Arabic, Czech, Russian, German, Spanish, Indonesian, Chinese |

Table 13: Few-shot accuracy on multilingual benchmarks for different pruning methods (Qwen3-8B, 70% sparsity). Bold values indicate the best performance.

| Accuracy↑ | Belebele | translated-ARC | translated-HellaSwag | Global-MMLU |
|---|---|---|---|---|
| Qwen3-8B | 0.8796 | 0.5499 | 0.4691 | 0.6777 |
| Mixed-data | 0.2331 | 0.2132 | 0.2523 | 0.2617 |
| Mixed-data + training | 0.2664 | 0.2311 | 0.2846 | 0.2695 |
| Lang-Prune | 0.2626 | 0.2169 | 0.2639 | **0.2716** |
| Lang-Prune + training | **0.2686** | **0.2468** | **0.2904** | 0.2646 |

Table 13 shows that both before and after post-training, mixed-data pruned models remain substantially below Lang-Prune in downstream accuracy, indicating that randomly mixing calibration data fails to preserve structures useful for adaptation. *translated-HellaSwag* exhibits the largest relative improvement after post-training, whereas *Global-MMLU* shows smaller or inconsistent gains, suggesting that tasks relying on broad general knowledge may be less sensitive to the structural differences preserved by Lang-Prune.

Overall, these results highlight the **task- and language-specific benefits** of Lang-Prune. Post-training recovery is most effective for tasks with high cross-lingual interference, confirming that preserving language-specific structures during pruning produces models that remain more adaptable across languages and tasks, even if improvements on general-knowledge-heavy benchmarks like Global-MMLU are limited.

## A.7 LANG-PRUNE GENERALITY ON WANDA

We conducted additional experiments to evaluate the behavior of Lang-Prune when applied to Wanda Sun et al. (2024), an unstructured pruning method that removes weights or rows/columns without preserving structured computational units. Under 30% sparsity, we compared three strategies: **Mixed-data Pruning** (mixed 9-language calibration), **Monolingual Pruning** (per-language calibration), and **Lang-Prune (max aggregation)** applied on top of Wanda.

Table 14: Perplexity (PPL) of Wanda under 30% sparsity. Lang-Prune shows poor performance, consistent with its reliance on structured pruning units.

| Wanda | Base Model | Mixed-data | Monolingual | Lang-Prune |
|---|---|---|---|---|
| **Avg. PPL↓** | 11.02 | 11.26 | 11.25 | 16.56 |

Lang-Prune performs poorly under Wanda, which is consistent with the design of multilingual importance estimation. Wanda operates at the level of individual weights or rows/columns, which do not correspond to coherent functional components such as MLP channels or attention heads. Preserving individual weights, even if important for a specific language, does not preserve the functional behavior of the network. In contrast, Lang-Prune is intended for settings where pruning is applied over semantically meaningful structures. Methods such as LLM-Pruner, head-pruning, channel-pruning, or block-level structured pruning expose units that correspond to functional submodules. In these cases, per-language importance can be translated into actionable preservation across languages.

Overall, these results demonstrate that Lang-Prune is compatible with any structured pruning method, but unstructured methods like Wanda fall outside this scope by design. Effective multi-

lingual pruning requires preserving importance at the level of functional components rather than individual weights, which explains the performance difference observed with Wanda.

## A.8 LANG-PRUNE PERFORMANCE ACROSS MODEL SCALES

We conducted additional experiments to evaluate *Lang-Prune* across a broader range of model sizes within the `Qwen3` family, spanning from 0.6B to 14B parameters. This includes both compact models with minimal redundancy and larger-scale LLMs with richer representational capacity.

Table 15: Perplexity (PPL)↓ across Qwen3 model sizes under 50% sparsity. Lower is better. Lang-Prune consistently performs best for mid-to-large models.

| Model Size | 0.6B | 1.7B | 4B | 8B | 14B |
|---|---|---|---|---|---|
| Base Model | 23.95 | 16.52 | 14.28 | 11.02 | 9.50 |
| Mixed-data | 52.38 | 80.96 | 40.72 | 310.69 | 555.36 |
| Monolingual | **47.59** | **36.47** | 32.72 | 36.72 | 69.10 |
| **Lang-Prune** | 56.07 | 45.68 | **31.43** | **24.30** | **31.60** |

Across mid-to-large model scales (4B, 8B, and 14B), Lang-Prune achieves the lowest perplexity among the pruning strategies. This indicates that modeling cross-lingual importance becomes increasingly beneficial as representational capacity grows and more structured redundancy exists in the network. At the smallest scale (0.6B), Lang-Prune performs worse than the other methods, likely due to the severely limited redundancy of tiny models; removing entire structured units under 50% sparsity substantially reduces capacity that these models cannot afford to lose. For the 1.7B model, the differences between methods are smaller and exhibit higher variance, suggesting a transitional regime where cross-lingual structure begins to emerge but remains fragile under structured pruning.

Overall, the results reveal a clear trend: the advantages of Lang-Prune amplify with increasing model size, demonstrating that language-sensitive aggregation generalizes across scales and is particularly effective for realistic multilingual deployment settings (4B parameters and above).

These findings also contextualize the relationship between pruning and training smaller dense models. In a zero-shot setting, a dense model trained from scratch at a given parameter scale (e.g., Qwen3-4B) typically achieves lower perplexity than a larger model pruned to the same effective size (e.g., 8B at 50% sparsity). This difference largely stems from data regimes: dense models are trained with full-scale curated corpora and task objectives, whereas one-shot pruning focuses purely on structural compression using minimal calibration data. Consequently, **pruned models should be viewed as strong initializations** that preserve the parent model's tokenizer, alignment, and behavioral characteristics, **rather than direct substitutes for fully trained dense models**. When further post-training is applied (Section 4.2), pruned variants recover substantial capability, supporting their practical role in adapting a single high-quality parent model to multiple deployment constraints.

