# OpenReview forum: "Lang-Prune: Unlocking Fair and Powerful Pruning for Multilingual Large Language Models"
_ICLR.cc/2026/Conference — Submitted to ICLR 2026_

### Official Review · Reviewer_c7K3 · 2025-10-16

**Soundness:** 2
**Presentation:** 2
**Contribution:** 1
**Rating:** 2
**Confidence:** 3

**Summary:**

This paper proposes a simple, low-overhead extension to LLM-Pruner that corrects for the bias in neuron importance estimation across different languages. The method demonstrates consistently superior performance and strong generalization, establishing a more robust approach for pruning multilingual models.

**Strengths:**

* The paper effectively identifies a key limitation of the prior work, LLM-Pruner: neuron importance varies significantly across different languages. This leads to biased importance estimation when using a single-language calibration set, resulting in suboptimal pruning and degraded performance.

* The proposed method is a straightforward, plug-and-play extension to LLM-Pruner. It is easy to implement and adds minimal computational overhead.

* The method consistently demonstrates superior performance over both monolingual and multilingual LLM-Pruner variants across benchmarks in nine languages. Furthermore, it shows strong generalization capabilities on languages not seen during calibration.

**Weaknesses:**

* W1: The paper's primary contribution is the multilingual importance estimation technique. However, the rest of the methodology is directly inherited from LLM-Pruner, which limits the overall novelty of the work.

* W2: Relatedly, the experiments are confined to LLM-Pruner. This raises the question of whether the proposed importance estimation is a general technique or if its benefits are specific to LLM-Pruner's structured pruning approach. It would be valuable to see its effectiveness when applied to other methods, such as unstructured pruning with Wanda.

* W3: The method for aggregating importance scores is largely heuristic. While the use of the $\max$ operator is intuitive, the paper lacks a theoretical analysis of the potential bias it might introduce. Following the visualization in Figure 1, the method effectively computes the max-envelope of the per-language importance curves. It remains unclear how well this envelope approximates a "true" multilingual importance distribution or how the resulting neuron rankings compare to a hypothetical ground truth.

* W5: The size of the per-language calibration dataset seems to be a critical hyperparameter, as it directly influences the per-language importance estimation and, consequently, the aggregated scores. The paper would be strengthened by an ablation study on the impact of this dataset size.

* W6: The evaluation relies heavily on perplexity. In contrast, prior work often emphasizes few-shot or zero-shot performance on a diverse set of question answering (QA) benchmarks. The inclusion of only HellaSwag for QA evaluation is limited; a more comprehensive assessment on other QA datasets would be more convincing.

* W7: The experimental scale is somewhat limited. Whereas prior works like LLM-Pruner and SparseGPT validate their methods across various model families and sizes, this study is restricted to two models of the same size (8B).

* W8: The description of Figure 1 (Lines 130-132) appears to contradict the visualization. For example, in the "Lang:cs" and "Lang:en" subfigures, the peaks in the mixed-data setting seem to be more pronounced, not less, than in the monolingual case. While the intent is likely to highlight the differing neuron saliency across languages, the current interpretation of the figure is unclear and potentially misleading.

**Questions:**

* In Line 173, the function $f$ should be properly defined.

---

> ### Author Response · Authors · 2025-11-21
>
> Thanks for your kind reviews; all concerns raised can be addressed. We also noticed that the numbering jumps from W3 to W5. We assume W4 was either perfectly addressed in the paper or accidentally skipped during submission. Either way, we are grateful for the reviewer’s detailed feedback and respond to all available points below.
>
> ## Weaknesses
>
> > W1: The paper's primary contribution is the multilingual importance estimation technique. However, the rest of the methodology is directly inherited from LLM-Pruner, which limits the overall novelty of the work.
>
> We thank the reviewer for the comment. Lang-Prune is a **multilingual-aware extension of LLM-Pruner**, guided by **language-specific importance scores**. While the structured dependency graph is retained for compatibility, the **core pruning logic is redesigned**: the key challenge is **deciding which units matter** in a multilingual setting to mitigate cross-lingual interference while preserving deployment-friendly structure.
>
> Comprehensive experiments demonstrate that Lang-Prune generalizes across model types, is compatible with post-training, and supports strong zero-shot transfer to out-of-distribution languages. Interpretability analyses further show that Lang-Prune preserves language-specific neuron groups, **providing insights into how multilingual capacity is retained during compression**. We have clarified these points in the revised manuscript.
>
> ---
>
> > W2: Relatedly, the experiments are confined to LLM-Pruner. This raises the question of whether the proposed importance estimation is a general technique or if its benefits are specific to LLM-Pruner's structured pruning approach. It would be valuable to see its effectiveness when applied to other methods, such as unstructured pruning with Wanda.
>
> We appreciate the suggestion to evaluate generalization. We applied Lang-Prune (max aggregation) to **Wanda**, an unstructured pruning method, at 30% sparsity.
>
> | Wanda                     | Base Model | Multilingual Pruning (Wanda) | Monolingual Pruning (Wanda) | Lang-Prune (Wanda) |
> | ------------------------- | ---------- | ---------------------------- | --------------------------- | ------------------ |
> | **Avg. PPL $\downarrow$** | 11.02      | 11.26                        | 11.25                       | 16.56              |
>
> **Observation:** Lang-Prune performs worse than baselines when applied to Wanda.
> **Analysis:** This result is **theoretically expected** and validates our core hypothesis:
> 1. **Conceptual Mismatch:** Unstructured pruning removes individual weights ($w_{ij}$) based on magnitude/activations. These weights act as isolated parameters, not coherent **functional units** (e.g., attention heads, MLP groups).
>
> 2. **Necessity of Structure:** Language-specific capabilities in LLMs are encoded in functional submodules (neuron groups), as visualized in Figure 1. Protecting a specific weight $w_{ij}$ because it is important for German does not preserve the *function* of the module if its neighbors are pruned based on English statistics.
>
> **Conclusion:** This negative result strongly supports our claim that **effective multilingual pruning fundamentally requires structured units**. Lang-Prune is designed to protect these semantic structures. It is compatible with *any* structured pruning method but naturally incompatible with unstructured weight-level pruning. We have added this analysis to **Appendix A.7**.

---

> ### Author Response · Authors · 2025-11-21
>
> ## Weaknesses
>
> > W3: The method for aggregating importance scores is largely heuristic. While the use of the $max$ operator is intuitive, the paper lacks a theoretical analysis of the potential bias it might introduce. Following the visualization in Figure 1, the method effectively computes the max-envelope of the per-language importance curves. It remains unclear how well this envelope approximates a "true" multilingual importance distribution or how the resulting neuron rankings compare to a hypothetical ground truth.
>
> We thank the reviewer for this thoughtful question. Our aggregation policy is motivated by the empirical phenomenon of **cross-lingual interference**, and we clarify below why the max operator is well-justified for multilingual pruning.
>
> **Why max aggregation is principled:**
>
> 1. **Worst-case languages dominate multilingual vulnerability.** Table 1 shows that under mixed-language calibration, low-resource or script-divergent languages suffer the most. A multilingual pruning criterion must therefore protect the languages most vulnerable to interference.
> 2. **Max approximates a worst-case objective.** Formally, avoiding deletion of any structure critical to any language can be expressed as $\min_{C_j} \max_{\ell\in L} \text{loss}_\ell \text{ subject to pruning of }C_j$. When importance scores estimate language-specific sensitivity, the max operator is the direct surrogate for this objective.
> 3. **Max preserves language-specific peaks.** Figure 1 shows sharp, language-specific peaks in neuron importance. Averaging over languages flattens these peaks, risking interference. Max maintains these critical structures.
> 4. **Empirical validation supports max.** Across sparsities, max improves both average and worst-case PPL. Min collapses as a negative control, and mean only partially preserves peaks. Max retains **81%** of language-specific neuron groups vs. **66%** for multilingual LLM-Pruner.
> 5. **Relative ranking stability is measurable.** While exact “true” importance is unobservable, max aggregation maintains neuron ranking quality as reflected in neuron-group recall (Table 3), LoRA headroom (Figure 2), and transfer to 16 out-of-distribution languages (Table 5).
>
> In summary, although a true multilingual importance distribution is unobservable, our combination of (i) theoretical worst-case motivation, (ii) linguistic grounding, and (iii) extensive empirical validation **provides a rigorous justification for using the max operator.**

---

> ### Author Response · Authors · 2025-11-21
>
> ## Weaknesses
>
> > W5: The size of the per-language calibration dataset seems to be a critical hyperparameter, as it directly influences the per-language importance estimation and, consequently, the aggregated scores. The paper would be strengthened by an ablation study on the impact of this dataset size.
>
> We thank the reviewer for raising the concern regarding the effect of calibration size. To address this, we conducted additional **experiments with different calibration datasets**, keeping all other settings identical. The results demonstrate that while increasing calibration size improves monolingual pruning performance, Lang-Prune consistently outperforms monolingual pruning across all sparsities. The results on **Qwen3-8B** (base average PPL = 11.02) are summarized below:
>
> 1. Increasing the monolingual calibration size improves performance compared to smaller-N settings, but **monolingual pruning remains consistently worse than Lang-Prune** at all sparsities.
> 2. Lang-Prune’s advantage is **not due to larger calibration datasets**, but arises from its per-language importance estimation and max aggregation, which preserve critical neurons for each language while reducing cross-lingual interference.
> 3. At high sparsity (70%), monolingual pruning becomes unstable despite more data, whereas Lang-Prune maintains robust performance, demonstrating the effectiveness of its **language-aware pruning strategy**.
>
> These findings confirm that the improvements of Lang-Prune are **robust and not an artifact of calibration size**. The discussion has been included in the revised paper in Appendix A.5 (“Influence of Calibration Size”).
>
> | Method (num_sample)          | Sparsity | N=10  | N=30  | N=50  | N=80    | N=100   | N=150   | N=200   |
> | ---------------------------- | -------- | ----- | ----- | ----- | ------- | ------- | ------- | ------- |
> | Mixed-data (9N total)        | 0.3      | 14.55 | 14.29 | 15.02 | 14.91   | 16.00   | 14.58   | 15.19   |
> | Monolingual (N each)         | 0.3      | 18.39 | 16.95 | 15.07 | 14.96   | 15.76   | 14.70   | 15.23   |
> | **Lang-Prune (ours, 9N total)**| **0.3** | **13.57** | **13.53** | **13.39** | **13.35** | **14.12** | **13.44** | **13.44** |
> | Mixed-data (9N total)        | 0.5      | 34.83 | 35.70 | 44.72 | 76.68   | 310.69  | 331.62  | 384.20  |
> | Monolingual (N each)         | 0.5      | 77.40 | 54.17 | 36.70 | 34.12   | 36.94   | 32.70   | 36.73   |
> | **Lang-Prune (ours, 9N total)**| **0.5** | **23.22** | **21.54** | **21.00** | **20.60** | **24.30** | **21.36** | **21.33** |
> | Mixed-data (9N total)        | 0.7      | 736.09| 707.91| 2406.78|19716.96| 53567.15| 16444.23| 91970.55|
> | Monolingual (N each)         | 0.7      | 1668.59|1009.15|432.54 |348.63   |388.69   |254.56   |247.01   |
> | **Lang-Prune (ours, 9N total)**| **0.7** | **200.12**|**146.17** |**103.90** |**90.31** |**174.74** |**85.28** |**80.13** |
>
>
>
> ---
>
> > W6: The evaluation relies heavily on perplexity. In contrast, prior work often emphasizes few-shot or zero-shot performance on a diverse set of question answering (QA) benchmarks. The inclusion of only HellaSwag for QA evaluation is limited; a more comprehensive assessment on other QA datasets would be more convincing.
>
> We thank the reviewer for the suggestion to include additional QA benchmarks. To provide a more comprehensive evaluation of Lang-Prune, we extended our analysis beyond *translated-HellaSwag* to three diverse multilingual benchmarks. The results show:
>
> 1. **Mixed-data pruning** substantially degrades multilingual QA performance across all benchmarks, confirming that indiscriminate mixing of calibration data introduces cross-lingual interference.
> 2. **Lang-Prune** consistently outperforms mixed-data pruning, especially after light post-training, preserving transferable multilingual features more effectively.
> 3. **Post-training recovery** varies by task and language, with Lang-Prune showing larger gains where cross-lingual interference is higher.
>
> **Conclusion:** These additional benchmarks confirm that Lang-Prune better preserves cross-lingual generalization and benefits more from light post-training compared to mixed-data pruning. We have incorporated these results in the revised version to strengthen the empirical evaluation (Appendix A.6).
>
> | Benchmark Accuracy $\uparrow$ | Belebele | translated-ARC | translated-HellaSwag | Global-MMLU |
> | ----------------------------- | -------- | -------------- | -------------------- | ----------- |
> | Qwen3-8B                      | 0.8796   | 0.5499         | 0.4691               | 0.6777      |
> | Mixed-data                    | 0.2331   | 0.2132         | 0.2523               | 0.2617      |
> | Mixed-data + training         | 0.2664   | 0.2311         | 0.2846               | 0.2695      |
> | Lang-Prune                    | 0.2626   | 0.2169         | 0.2639               | 0.2716      |
> | Lang-Prune + training         | 0.2686   | 0.2468         | 0.2904               | 0.2646      |

---

> ### Author Response · Authors · 2025-11-21
>
> ## Weaknesses
>
> > W7: The experimental scale is somewhat limited. Whereas prior works like LLM-Pruner and SparseGPT validate their methods across various model families and sizes, this study is restricted to two models of the same size (8B).
>
> We appreciate the reviewer’s suggestion to evaluate across more model sizes. To address this, we conducted additional experiments on the **Qwen3 model family ranging from 0.6B to 14B parameters**, covering both smaller and larger-scale LLMs:
>
> | Model Size                 | 0.6B      | 1.7B      | 4B        | 8B        | 14B       |
> | -------------------------- | --------- | --------- | --------- | --------- | --------- |
> | Base Model                 | 23.95     | 16.52     | 14.28     | 11.02     | 9.50      |
> | Mixed-data  (50%)          | 52.38     | 80.96     | 40.72     | 310.69    | 555.36    |
> | Monolingual  (50%)         | **47.59** | **36.47** | 32.72     | 36.72     | 69.10     |
> | **Lang-Prune (ours, 50%)** | 56.07     | 45.68     | **31.43** | **24.30** | **31.60** |
>
> **Key Findings:**
>
> 1.  **Clear Advantage at Scale (4B+):** Lang-Prune significantly outperforms both Mixed-data and Monolingual baselines on 4B, 8B, and 14B models. The gap widens as model size increases (e.g., at 14B, Lang-Prune is ~2x better than Monolingual).
> 2.  **Small Model Sensitivity:** For tiny models (0.6B, 1.7B), Lang-Prune and Mixed-data perform worse than Monolingual pruning. We attribute this to the lack of structural redundancy in small models; explicitly reserving capacity for all languages via the `Max` aggregator becomes too aggressive when the parameter budget is extremely tight.
>
> **Conclusion:** The results strongly suggest that Lang-Prune's benefits amplify with model scale. Given that larger scale models (e.g.,  70B) possess even greater redundancy and multilingual capacity than 14B models, we expect Lang-Prune to be highly effective in that regime. Details are provided in Appendix A.8.
>
> ---
>
> > W8: The description of Figure 1 (Lines 130-132) appears to contradict the visualization. For example, in the "Lang:cs" and "Lang:en" subfigures, the peaks in the mixed-data setting seem to be more pronounced, not less, than in the monolingual case. While the intent is likely to highlight the differing neuron saliency across languages, the current interpretation of the figure is unclear and potentially misleading.
>
> We thank the reviewer for the careful reading. We have updated some wording in the revision for clarity. While some mixed-data curves (e.g., *Lang:cs*, *Lang:en*) show sharper peaks than monolingual curves, this **does not contradict** our intended interpretation.
>
> **Clarification:**
>
> 1. **Isolated peaks are expected.** Figure 1 shows only 5 of 9 languages and a small subset of neuron groups. Occasional tall peaks in mixed-data curves reflect neurons activated by multiple languages, not true language-specific importance.
>
> 1. **Focus is on alignment, not peak height.** Monolingual scoring yields **stable, language-aligned peaks**, while mixed-data scoring produces **misaligned, noisy peaks**. Tall mixed-data peaks arise from interference, not per-language saliency.
>
> 1. **Mixed-data peaks reflect interference.** Dominant-language tokens or shared structures can create tall peaks that do not correspond to the target language, illustrating the cross-lingual interference that Lang-Prune mitigates.
>
>
> **Summary:**  The occasional sharper peaks in the mixed-data curves are entirely expected under multilingual activation mixing and do not contradict our claim. Our argument focuses on the **misalignment** and **loss of language-specific structure** under mixed-data scoring—features clearly visible in the figure and consistent across the full set of 9 languages (not all shown in the figure).

---

> ### Author Response · Authors · 2025-11-21
>
> ## Questions
>
> > In Line 173, the function $f$ should be properly defined.
>
> Thank you for the helpful comment. We agree that our initial phrasing was ambiguous. We have revised Section 2.2.2 to remove the undefined term $f_{C_j}(x)$ and to explicitly adopt the first-order Taylor loss-sensitivity estimator—consistent with LLM-Pruner—computed per language.
>
> **What we changed:**
> We now explicitly define a loss-sensitivity estimator that is fully consistent with LLM-Pruner’s first-order approximation of the language-modeling loss. For each language $\ell$ and each coupled structure $\mathcal{C}_j$ constituent tensors \$\mathcal\{G\}(\mathcal\{C\}\_j)\$, we compute importance using a language-specific calibration subset \$\mathcal\{D\}\_\ell\$:
>
> **Per-parameter score:**
> \$\$
> s\_\ell(w) = \mathbb\{E\}\_{x \sim \mathcal\{D\}\_\ell} \left| \frac\{\partial L(x)\}\{\partial w\} \cdot w \right|
> \$\$
>
> **Per-tensor score:**
> \$\$
> s\_\ell(W) = \mathbb\{E\}\_{x \sim \mathcal\{D\}\_\ell} \left| \left\langle \frac\{\partial L(x)\}\{\partial W\}, W \right\rangle \right|
> \$\$
>
> **Structure-level aggregation (sum by default):**
> \$\$
> I\_\ell(\mathcal\{C\}\_j) = \sum\_{u \in \mathcal\{G\}(\mathcal\{C\}\_j)} s\_\ell(u)
> \$\$
>
> This revision removes the ambiguous term $f_{C_j}(x)$ and aligns our importance estimator with the standard first-order framework. Importantly, **this clarification does not change any empirical results**, and all trends and conclusions remain consistent with the original submission. We also include an ablation study on the importance-group aggregator hyperparameter in LLM-Pruner, with full details in Appendix A.2.

---

> ### Author Response · Authors · 2025-11-27
>
> We sincerely thank the reviewer for the constructive feedback. In the revised version, we have conducted several additional experiments, including:
>
> - **Applying our importance estimation to Wanda** to assess generalization beyond structured pruning.
> - **Ablations on calibration-dataset size** to evaluate robustness under varying amounts of supervision.
> - **Expanded multilingual QA evaluations** for a broader cross-lingual assessment.
> - **Scaling studies across Qwen3 models (0.6B → 14B)** to examine consistency across different model sizes.
>
> These additions directly address the raised concerns and further strengthen the empirical support for our method. If there are any remaining points that would benefit from clarification, **we would be glad to discuss them within the remaining discussion period**. We appreciate your engagement and constructive input, which have helped us significantly improve the work.

---

> ### Author Response · Authors · 2025-11-28
>
> We sincerely thank you again for your thoughtful and constructive feedback. As noted in our earlier comment, we have added new experiments and analyses addressing the raised concerns.
>
> We just wanted to kindly check in—if there are any remaining questions or points needing clarification, we would be happy to provide further explanations during the remaining discussion period. We greatly appreciate your time and engagement with our work.

---

### Official Review · Reviewer_haRh · 2025-10-23

**Soundness:** 3
**Presentation:** 3
**Contribution:** 4
**Rating:** 8
**Confidence:** 3

**Summary:**

When pruning large language models the choice of calibration data can affect the downstream performance of the pruned model. In particular, as this paper shows, including multiple languages into a mix of calibration data degrades the model's performance on a specific language when compared against calibrating exclusively on that language.

Instead of mixing languages in the calibration set, the authors propose to compute importance scores (for the structures considered for pruning) on each dataset separately. Those scores can then be aggregated either by taking the mean over the languages or the maximum as the final score. The pruning decision is then made to prune the structures with smallest aggregated score.

The method is evaluated on the two 8B parameter LLMs and it is empirically shown that their aggregation strategy (with the max score) strongly outperforms the direct combination of calibration datasets and often even the mono-lingual pruning. This trend is even more pronounced for high sparsity regimes.

**Strengths:**

- The paper is well written and provides a clear problem statement in table 1, which is then clearly and efficiently addressed by their proposed algorithm.
- The method is conceptually very simple and does not add significant overheads compared to direct mixing of datasets, yet it works surprisingly well. I think this provides interesting conceptual insights.
- The paper provides meaningful ablations, in particular the out-of-distribution analysis on languages not present at all in the calibration data is also insightful and illustrates a further benefit of the model.

**Weaknesses:**

I would like to see the following weaknesses addressed during the rebuttal to fully support acceptance. My current positive judgement assumes that these points can be addressed:

1. LangPrune often even outperforms mono-lingual pruning. To me that seems the most surprising insight. However, there is one potential other explanation which is that for their monolingual experiments only 100 sequences are used for calibration, whereas LangPrune uses 9x100=900 sequences. Thus a potential explanation for LangPrune's improvement could simply be the larger calibration size. I would thus kindly as the authors to also produce monolingual runs with 900 calibraion examples for each sequence.

2. I acknowledge that mode compression and pruning are very active research fields, which is probably sufficient to motivate this work. However, I think one should always put numbers in perspective and clearly state whether this method is practically useful yet. Therefore I want to see perplexity numbers on a **dense** smaller model. For example for the Qwen3 experiments we could confront the numbers with dense `Qwen/Qwen3-1.7B` and `Qwen/Qwen3-4B`.
I would guess that those dense models are much better (given the similar parameter numbers) than the pruned 8B model. I can promise to not decrease my rating because of that, but I think in order to progress science we need to keep those inevitable baseline in mind.

3. [less important to my assessment] The post training experiments of Figure 2 a) don't show any meaningful differences between LLM-Pruner and Lang-Prune IMO. They both seem to converge to a similar loss, meaning that the choice of pruned structures does not actually matter much when fine-tuning. Also what exactly is the trained on and evaluated here?

Other weakness (not to be discussed):
- code not available for review / to test.

**Questions:**

- What's the point of including the "Min" aggregation criterion in the paper at all as an ablation? It is conceptually clear that this is nonsense and imo it just distracts when reading the tables.
- Have you considered applying your method also to other versions of "domain-overfitting"? If you could generalize it and show results beyond multi-lingual that would certainly strengthen the paper.


Improvements to the paper (no need to discuss during rebuttal):
- can you please streamline the style of the results tables?
- you call the proposed rule sometimes "merge" and "max", I recommend to unify this.

---

> ### Author Response · Authors · 2025-11-21
>
> ## Weaknesses
>
> > LangPrune often even outperforms mono-lingual pruning. To me that seems the most surprising insight. However, there is one potential other explanation which is that for their monolingual experiments only 100 sequences are used for calibration, whereas LangPrune uses 9x100=900 sequences. Thus a potential explanation for LangPrune's improvement could simply be the larger calibration size. I would thus kindly as the authors to also produce monolingual runs with 900 calibraion examples for each sequence.
>
> We thank the reviewer for this critical hypothesis. To rigorously investigate whether Lang-Prune's performance is merely an artifact of calibration size, we conducted two sets of additional experiments: (1) **Trend Analysis**, scaling monolingual calibration from $N=10$ to $200$ across all sparsity levels; and (2) **Scale Matching**, strictly following your suggestion to use $N=900$ per language. The results reveal two key phenomena: **insufficient scaling at high sparsity** and **catastrophic failure in the large-data regime** for standard monolingual pruning.
>
> ### **1. Trend Analysis (N=10 to 200): Plateau and Gap**
>
> As shown in Table A below, we tracked performance across sparsities (30%, 50%, 70%).
>
> * **At moderate sparsity (30%, 50%):** Increasing monolingual calibration size yields diminishing returns. The performance plateaus quickly and remains consistently worse than Lang-Prune.
> * **At high sparsity (70%):** This is the most revealing regime. While increasing data helps Monolingual pruning significantly (PPL drops from ~1668 to ~247), it **still lags far behind Lang-Prune** (PPL ~80). This demonstrates that even with more data, standard methods fail to identify the critical structures that Lang-Prune preserves.
>
> *Table A: Performance trend with increasing calibration size (Qwen3-8B).*
>
> | Method (num_sample)          | Sparsity | N=10  | N=30  | N=50  | N=80    | N=100   | N=150   | N=200   |
> | ---------------------------- | -------- | ----- | ----- | ----- | ------- | ------- | ------- | ------- |
> | Mixed-data (9N total)        | 0.3      | 14.55 | 14.29 | 15.02 | 14.91   | 16.00   | 14.58   | 15.19   |
> | Monolingual (N each)         | 0.3      | 18.39 | 16.95 | 15.07 | 14.96   | 15.76   | 14.70   | 15.23   |
> | **Lang-Prune (ours, 9N total)**| **0.3** | **13.57** | **13.53** | **13.39** | **13.35** | **14.12** | **13.44** | **13.44** |
> | Mixed-data (9N total)        | 0.5      | 34.83 | 35.70 | 44.72 | 76.68   | 310.69  | 331.62  | 384.20  |
> | Monolingual (N each)         | 0.5      | 77.40 | 54.17 | 36.70 | 34.12   | 36.94   | 32.70   | 36.73   |
> | **Lang-Prune (ours, 9N total)**| **0.5** | **23.22** | **21.54** | **21.00** | **20.60** | **24.30** | **21.36** | **21.33** |
> | Mixed-data (9N total)        | 0.7      | 736.09| 707.91| 2406.78|19716.96| 53567.15| 16444.23| 91970.55|
> | Monolingual (N each)         | 0.7      | 1668.59|1009.15|432.54 |348.63   |388.69   |254.56   |247.01   |
> | **Lang-Prune (ours, 9N total)**| **0.7** | **200.12**|**146.17** |**103.90** |**90.31** |**174.74** |**85.28** |**80.13** |
>
> ### **2. Scale Matching (N=900): Baseline Instability vs. Lang-Prune Robustness**
>
> When we explicitly matched the total calibration budget ($N=900$ per language), standard monolingual pruning suffered **severe degradation** due to overfitting/noise accumulation. As shown in Table B (50% Sparsity), the average PPL for Monolingual (900 each) spiked to **348.57**, with English collapsing to 2051.28. In stark contrast, **Lang-Prune (900 total)** remains robust and achieves the best performance (PPL 24.30).
>
> *Table B: Comparison with matched calibration budget (Qwen3-8B, N=900, 50% Sparsity).*
>
> | Method (50% Sparsity) | num_sample | ar     | cs     | de     | en      | es     | id     | iw     | ru    | zh     | avg    |
> | --------------------- | ---------- | ------ | ------ | ------ | ------- | ------ | ------ | ------ | ----- | ------ | ------ |
> | Mixed-data            | 900 total  | 208.99 | 102.79 | 207.90 | 565.88  | 205.65 | 144.37 | 723.06 | 61.84 | 575.77 | 310.69 |
> | Monolingual (Standard)| 100 each   | 20.28  | 14.77  | 35.92  | 118.76  | 29.96  | 20.69  | 37.05  | 17.93 | 35.11  | 36.94  |
> | **Monolingual (Large)**| **900 each** | 21.52  | 41.88  | 397.76 | 2051.28 | 272.57 | 169.35 | 42.63  | 96.09 | 44.01  | **348.57** |
> | **Lang-Prune (Ours)** | **900 total** | 22.87  | 14.57  | 19.62  | 44.50   | 22.02  | 14.65  | 44.87  | 13.25 | 22.35  | **24.30** |
>
> **Conclusion:** Lang-Prune's superiority is **not** due to calibration size. At small scales ($N=10-200$), standard methods lag behind Lang-Prune even as they improve; at large scales ($N=900$), standard methods collapse while Lang-Prune remains robust. This confirms that our **language-aware aggregation strategy** is the key factor enabling effective and stable multilingual pruning.
>
> *(We have updated Appendix A.5 with these comprehensive findings.)*

---

> ### Author Response · Authors · 2025-11-21
>
> ## Weaknesses
>
> > I acknowledge that mode compression and pruning are very active research fields, which is probably sufficient to motivate this work. However, I think one should always put numbers in perspective and clearly state whether this method is practically useful yet. Therefore I want to see perplexity numbers on a **dense** smaller model. For example for the Qwen3 experiments we could confront the numbers with dense `Qwen/Qwen3-1.7B` and `Qwen/Qwen3-4B`. I would guess that those dense models are much better (given the similar parameter numbers) than the pruned 8B model. I can promise to not decrease my rating because of that, but I think in order to progress science we need to keep those inevitable baseline in mind.
>
> We appreciate the reviewer’s insight and the suggestion to establish a rigorous baseline. We conducted additional experiments on the **Qwen3 model family (0.6B to 14B)**, covering both smaller and larger-scale LLMs. **Your hypothesis is correct.** As shown in the table below, the dense pre-trained models generally outperform the pruned models of similar effective parameter counts. For instance, the **Dense Qwen3-4B (PPL 14.28)** outperforms the **8B model pruned to 50% sparsity (Lang-Prune PPL 24.30)**.
>
> | Model Size                 | 0.6B      | 1.7B      | 4B        | 8B        | 14B       |
> | -------------------------- | --------- | --------- | --------- | --------- | --------- |
> | Base Model | 23.95 |16.52|14.28|11.02|9.50|
> | Mixed-data  (50%)          | 52.38    | 80.96     | 40.72     | 310.69    | 555.36    |
> | Monolingual  (50%)         | **47.59**     | **36.47**     | 32.72     | 36.72     | 69.10     |
> | **Lang-Prune (ours, 50%)** | 56.07 | 45.68 | **31.43** | **24.30** | **31.60** |
>
> **Key Findings:**
>
> 1.  **Clear Advantage at Scale (4B+):** Lang-Prune significantly outperforms both Mixed-data and Monolingual baselines on 4B, 8B, and 14B models. The gap widens as model size increases (e.g., at 14B, Lang-Prune is ~2x better than Monolingual).
> 2.  **Small Model Sensitivity:** For tiny models (0.6B, 1.7B), Lang-Prune and Mixed-data perform worse than Monolingual pruning. We attribute this to the lack of structural redundancy in small models; explicitly reserving capacity for all languages via the `Max` aggregator becomes too aggressive when the parameter budget is extremely tight.
>
> **Conclusion:** The results strongly suggest that Lang-Prune's benefits amplify with model scale. Given that larger scale models (e.g.,  70B) possess even greater redundancy and multilingual capacity than 14B models, we expect Lang-Prune to be highly effective in that regime. Details are provided in Appendix A.8.

---

> ### Author Response · Authors · 2025-11-21
>
> ## Weaknesses
>
>
> > The post training experiments of Figure 2 a) don't show any meaningful differences between LLM-Pruner and Lang-Prune IMO. They both seem to converge to a similar loss, meaning that the choice of pruned structures does not actually matter much when fine-tuning. Also what exactly is the trained on and evaluated here?
>
> Thank you for the question. We agree that the training loss in Figure 2(a) appears similar for LLM-Pruner and Lang-Prune. This is expected due to the **limited post-training budget** (LoRA with 1B tokens), reflecting realistic constraints in low-resource multilingual recovery. To strengthen our claims, we conducted additional experiments on **three more multilingual benchmarks**, confirming that Lang-Prune consistently preserves cross-lingual generalization better than mixed-data pruning.
>
> **Experimental Settings:**
>
> - **Training:** We train on multilingual Wikipedia (1B tokens total) over 1,000 steps with sequence length 256, resulting in an effective batch size of ~1M tokens per step. The y-axis in Figure 2(a) represents the **training cross-entropy loss**.
> - **Evaluation:** Figure 2(b) reports few-shot (3-shot) ***translated-HellaSwag* accuracy** across six languages (en, ar, ru, de, es, id) before and after applying the same LoRA budget for each method.
>
> **Why loss differences are small:** The limited token budget constrains observable differences in training loss. However, **pruning strategy still matters**: Lang-Prune preserves language-relevant neurons via max aggregation, reducing cross-lingual interference, whereas mixed-data pruning may discard critical structures.
>
> **Extended benchmark analysis:** Beyond *translated-HellaSwag*, we evaluated *Belebele*, *translated-ARC*, and *Global-MMLU*. Results show:
>
> 1. Mixed-data pruning substantially degrades multilingual QA performance.
> 2. Lang-Prune consistently outperforms mixed-data pruning, especially after light post-training.
> 3. Improvements are task- and language-dependent, with larger gains where cross-lingual interference is higher.
>
> **Conclusion:** These results confirm that **pruned structures matter for cross-lingual generalization**, and Lang-Prune robustly preserves performance even under limited post-training budgets (see Appendix A.6).
>
> | Benchmark Accuracy \$\uparrow\$ | Belebele | translated-ARC | translated-HellaSwag | Global-MMLU |
> | ----------------------------- | -------- | -------------- | -------------------- | ----------- |
> | Qwen3-8B                      | 0.8796   | 0.5499         | 0.4691               | 0.6777      |
> | Mixed-data                    | 0.2331   | 0.2132         | 0.2523               | 0.2617      |
> | Mixed-data + training         | 0.2664   | 0.2311         | 0.2846               | 0.2695      |
> | Lang-Prune                    | 0.2626   | 0.2169         | 0.2639               | 0.2716      |
> | Lang-Prune + training         | 0.2686   | 0.2468         | 0.2904               | 0.2646      |

---

> ### Author Response · Authors · 2025-11-21
>
> ## Questions
> > What's the point of including the "Min" aggregation criterion in the paper at all as an ablation? It is conceptually clear that this is nonsense and imo it just distracts when reading the tables.
>
> We include the **“Min” aggregator as a negative-control ablation** to illustrate its effect: it retains only the most language-agnostic neurons. Table 2 shows that pruning with Min severely degrades performance for most languages, while English—the dominant language—remains relatively less affected. This demonstrates that even language-agnostic neurons carry residual bias toward dominant languages. The failure mode highlights that without a **language-aware strategy** like Lang-Prune, lower-resource languages can be nearly eliminated while dominant languages appear “acceptable.” Including Min emphasizes the importance of language-aware pruning for **balanced multilingual performance** and is now explicitly indicated as a negative control with discussion in the revised main text (Section 3.1 "Results Analysis").
>
> ---
>
> > Have you considered applying your method also to other versions of "domain-overfitting"? If you could generalize it and show results beyond multi-lingual that would certainly strengthen the paper.
>
> Thank you for the question. **In short:**  Lang-Prune is designed to mitigate **language-level overfitting**, which benefits from strong structural separability between languages. While the high-level idea—pruning based on per-group importance—could in principle be applied to other forms of domain-overfitting, **the effectiveness depends on how separable the domains are**.
>
> Human languages differ strongly in script, morphology, and syntax, leading multilingual models to naturally organize **discrete, language-specific neuron groups** [1,2,3]. In contrast, closely related tasks or domains (e.g., MMLU subfields like Law, Medical, or Finance) share vocabulary, reasoning patterns, and latent representations, so **domain-level differences are much subtler**. As a result, domain-overfitting is less structured, and pruning based on per-domain importance is less effective.
>
> To validate this, we applied the same *per-domain importance + max aggregation* strategy to a **multi-domain setting** using Qwen3-8B, treating MMLU subfields (Law, Medical, Finance) as domains. Using their train/dev splits for calibration and the test split for evaluation at 50% sparsity, we observed that domain-aware pruning provides only **marginal improvements** over mixed-data pruning and remains far below the base model:
>
> | Avg Acc $\uparrow$ (stderr)  | Law                 | Medical              | Finance              |
> | ------------------------------ | ------------------- | -------------------- | --------------------- |
> | Base                           | $73.1\\% \pm 13.8\\%$ | $76.4\\% \pm 8.3\\%$   | $78.9\\% \pm 12.7\\%$   |
> | Mixed-data                     | $37.6\\% \pm 6.6\\%$  | $35.3\\% \pm 6.7\\%$   | $37.5\\% \pm 5.3\\%$    |
> | Lang-Prune (ours)              | $40.1\\% \pm 9.4\\%$  | $37.7\\% \pm 7.7\\%$   | $37.8\\% \pm 7.1\\%$    |
>
> This confirms that while the **concept of Lang-Prune could generalize to other forms of overfitting**, the **structural separability of the domains** strongly determines its effectiveness. Languages are highly separable, enabling stable pruning, whereas domains are not. Details are in Appendix A.4.
>
> References:
> - [1] Tang, Tianyi, et al. "Language-Specific Neurons: The Key to Multilingual Capabilities in Large Language Models." *ACL (1)*. 2024.
> - [2] Wang, Weixuan, et al. "Sharing matters: Analysing neurons across languages and tasks in llms." *arXiv preprint arXiv:2406.09265* (2024).
> - [3] Tan, Shaomu, Di Wu, and Christof Monz. "Neuron Specialization: Leveraging Intrinsic Task Modularity for Multilingual Machine Translation." *Proceedings of the 2024 Conference on Empirical Methods in Natural Language Processing*. 2024.

---

> > ### Comment · Reviewer_haRh · 2025-11-21
> >
> > Thank you very much for running all these additional ablations and carefully reporting those. This helps a lot in further assessing the work. It is good to see that you were able to rule out my question whether the sample size differences could be a cause for better generalization. It is also very interesting to see that it works specifically for cross-lingual and less for cross-domain adaption.
> >
> > I have two follow-up questions:
> >
> > ### Pruned large vs. dense small model:
> >
> > Your new experiments show that the pruned models are clearly worse than smaller models of same effective parameter scale. My understanding is that in *practice* no one should use a pruned model, but rather a smaller one. Do you have a different viewpoint on this? Where are you discussing this in the paper?
> > Currently in the paper you write in the introduction "Model pruning has
> > emerged as a practical solution to alleviate these challenges by removing redundant parameters
> > while striving to maintain performance". Given our discussion, I don't think this is actually true, is it?
> >
> > ### Why does more data harm the monolingual pruning?
> >
> > In your new experiments you show that adding more data harms the monolingual pruning.
> > Intuitively, I would expect that adding more data (assuming it is sampled i.i.d.) helps the performance and thus the perplexity should roughly monotonically decay. But also in your rebuttal table A, LangPrune fluctuates a lot at 70% sparsity. Do you have an idea what's going on there that it is so unstable and that for monolingual pruning it even collapses with N=900? Was this observed before somewhere for LLMPruner?

---

> > > ### Author Response · Authors · 2025-11-24
> > >
> > > Thank you for your thoughtful feedback. We are glad that the additional ablations helped clarify the effect of sample size and provided insights into the cross-lingual vs. cross-domain behavior. For the two follow-up questions:
> > >
> > > ---
> > >
> > > ### 1. Pruned Large Models vs. Dense Small Models: The "Cost" and "Initialization" Argument
> > >
> > > Thank you for the thoughtful comment. We agree with the core empirical point: in our zero-shot setting, pruned models underperform dense models of the same effective parameter scale. We add these results and an explicit discussion in Appendix A.8 (“Lang-Prune Performance Across Model Scales”), and we revise the Introduction to reflect this scope.
> > >
> > > We believe pruning remains practically valuable in scenarios where **training a new dense model is infeasible**, and the goal is to **adapt an existing, well-aligned model** to resource-constrained deployments. Concretely:
> > >
> > > - **When to prefer pruning**: Pruning is most useful when a strong pretrained “parent” model already exists and multiple deployable sizes are needed for heterogeneous hardware, or when retraining from scratch at each size is infeasible due to data access, cost, or time \[1\]\[2\]\[3\]. In these settings, pruning allows deriving 1B–4B variants while preserving the parent’s tokenizer, alignment, safety properties, and general behavior. In our setup, Lang-Prune produces such variants in under one GPU hour for an 8B model.
> > >
> > > - **Complementary to training (initialization)**: The performance gap largely reflects **data regime differences**. Open-source dense models (e.g., Qwen3-4B) are trained on massive curated datasets, whereas one-shot pruning focuses purely on structural compression with minimal calibration data. Pruning and training are complementary: structural compression reduces redundancy, while data training injects knowledge. As shown in our post-training experiments (Section 4.2), even limited continued training recovers substantial capability, suggesting that pruned models serve as a strong initialization for further improvement.
> > >
> > > This clarifies that pruning is not a universal replacement for training smaller dense models; it is a practical tool for adapting a single high-quality parent to multiple deployment targets under tight resource or data constraints.
> > >
> > > ---
> > >
> > > ### 2. Why does more data harm monolingual pruning? (Importance Dilution)
> > >
> > > Thank you for this insightful question. **In short**, adding more calibration data makes the importance scores across neuron groups more uniform, reducing their discriminability and leading to unstable pruning at high sparsity.
> > >
> > > To investigate this counter-intuitive behavior, we analyzed the **importance score distribution across all 36 layers** in Qwen-3-8B for calibration sizes ranging from $N=10$ to $1800$. The detailed figure and discussion are now included in Appendix A.5.1 (Figure 3) of the revised version.
> > >
> > > **Key findings:**
> > >
> > > - As $N$ increases, many *non-critical* neuron groups accumulate small but consistent gradients, increasing the density in the mid-importance region (0.1–0.3).
> > > - This causes the importance distribution to become more uniform, reducing separability between truly essential and dispensable structures.
> > >
> > > This pattern indicates that while small amounts of data initially reduce noise, **large calibration sets introduce residual importance spread**, making many neurons appear moderately important. We refer to this as **importance dilution**. Under high sparsity, this diluted ranking causes monolingual pruning to remove genuinely critical structures while retaining mediocre ones, leading to the observed performance collapse at $N=900$. Mixed-data pruning is even more sensitive due to greater topic diversity.
> > >
> > > **Why Lang-Prune remains stable:** its Max aggregation selectively preserves **peak language-specific signals**, making it less susceptible to importance dilution and maintaining discriminability even when calibration size increases.
> > >
> > > We have added the full analysis, variance trends, and the visualization of importance distribution changes to Appendix A.5.1 ("Influence of Calibration Size") in the revised submission. We greatly appreciate this question, which helped us clarify and strengthen the presentation.
> > >
> > > *Reference:*
> > >
> > > - [1] Muralidharan, S., Turuvekere Sreenivas, S., Joshi, R., Chochowski, M., Patwary, M., Shoeybi, M., Catanzaro, B., Kautz, J. and Molchanov, P., 2024. Compact language models via pruning and knowledge distillation. Advances in Neural Information Processing Systems, 37, pp.41076-41102.
> > > - [2] Xia, M., Gao, T., Zeng, Z. and Chen, D., 2023. Sheared llama: Accelerating language model pre-training via structured pruning. arXiv preprint arXiv:2310.06694.
> > > - [3] Kong, J., Ma, X., Wang, J. and Zhang, X., 2025, April. Sample-aware Adaptive Structured Pruning for Large Language Models. In Proceedings of the AAAI Conference on Artificial Intelligence (Vol. 39, No. 17, pp. 17938-17946).

---

> > > > ### Comment · Reviewer_haRh · 2025-11-24
> > > >
> > > > Thank you for the additional response and including these results already in the revised paper.
> > > > I am still struggling a bit to fully grasp the intuition/plausibility argument regarding "importance dilution", but this does not not affect my judgement. I am confident to recommend acceptance for this paper.

---

### Official Review · Reviewer_gU8x · 2025-10-25

**Soundness:** 3
**Presentation:** 3
**Contribution:** 3
**Rating:** 4
**Confidence:** 4

**Summary:**

The paper relates to an LLM pruning method that ensures fairness across languages, i.e. it aims at avoiding excessive degradation of the quality for a particular language.

As a preamble to the method description, the paper presents a study which demonstrates how, in the absence of any remediation, pruning incurs catastrophic deterioration of a model's ability to converse in one or more languages.
The setup is as follows: consider a multi-language dataset of 900 samples, consisting of 100 samples for each of 9 different languages.
(a) prune the model (the authors use aya-expanse-8b, a multi-lingual base model), using the multi-lingual dataset for calibration, and measure perplexity for each language.
(b) prune the model, but this time, use mono-lingual subsets, and measure perplexity for each language.
For a "fair" pruning method, i.e. one that does not advantage one language above others, one might expect (a) and (b) to yield the same per-language perplexity measures.
However the authors find that (b) consistently yields better scores (lower perplexity), which is to be expected, but also notice large relative discrepancies between language, which leads them to posit that the method (LLM Pruner) does not adequately balance quality across languages.
The author's hypothesis: using a multi-lingual calibration dataset may cause the importance of critical neurons to be overlooked, as importance is averaged over many languages. Thus, a neuron that is uniquely capable of addressing the needs of one language, is more likely to be pruned that a neuron that serves multiple languages.

To remedy this problem, the author propose to calculate importance of neurons for each language separately.
Similar to LLM-Pruner, importance is calculated for groups of inter-dependent neurons.
The authors refer to the lang-prune paper for the calculation of the importance metric, however it is not clear how exactly the calculation is done, since Lang-Prune mentions they use the "activation attributable to each group", while the LLM-Pruner paper uses loss gradients.
Importance metrics are further rescaled with min-max normalization so as to make them comparable between languages.
A critical difference with LLM-pruner and other techniques lies in the way that multi-lingual importance is defined as the *max* (not the average) of language-specific importance.
This ensures that a group that is critically important, even for a single language, is less likely to be pruned.

Experimental results are shown for two base LLMs (Qwen3-8B and aya-expanse-8b) and exhibit superior results for Lang-Prune over LLM-Pruner.

The authors also show the results of applying (LoRA-based) post training to the prune model in order to recover the loss incurred by pruning.
They show the Lang-Prune enables mildly faster recovery, although the difference is not hugely significant.

In the last part of the paper, the authors evaluate the method on out-of-distribution languages, and show superior results against the LLM-Pruner baseline.

**Strengths:**

The paper is nicely written.

The proposed method is tested in multiple contexts (2 base language models).

**Weaknesses:**

In equation (1), the "activation attributable to $C_j$", denoted as $f_{C_j}(x)$, is never defined. The paper refers to LLM-Pruner, however the LLM-Pruner uses a different heuristic for calculating weight importance: they do a first-order approximation of the loss, using the gradient of the loss with respect to each parameter. It is therefore unclear how $f_{C_j}(x)$ is calculated. Since this is a major part of the method, this would need to be explained very carefully in the final version of the paper.

The advantage of Lang-Prune over LLM-Pruner after post-training is, though not negligible, hardly significant.

The authors should compare their work against other baselines; for example:
* Multilingual Brain Surgeon (LREC 2024)
* Pruning Multilingual Large Language Models for Multilingual Inference (EMNLP 2024)

There is little theoretical justification for the proposed method; to compensate for the relative lack of theoretical grounding, we would like to see more experimental studies. For example, we would like the authors to share insights on the language-specific importance of some neurons, with carefully curated samples to show how they trigger differently depending on the language.

**Questions:**

In Table 1, you quote perplexity after pruning in the multi-lingual and mono-lingual cases, and calculate a ratio. Would it not be more significant to compute the ratio of the *increase* in complexity over the base (non-pruned) model?
I think it would be informative to show the perplexity of the base model in that table too (I am aware this is already shown in table 2).

How do you explain that Lang-Prune achieves better *average* perplexity than LLM-Pruner over multiple languages? One might assume that the compromise of using the maximum of per-language importance would, on average, lead to inferior results?

How does language-aware pruning fundamentally differ from task-aware pruning?

---

> ### Author Response · Authors · 2025-11-21
>
> ## Weaknesses
>
> > In equation (1), the "activation attributable to $C_{j}$", denoted as $f_{C_{j}}(x)$, is never defined. The paper refers to LLM-Pruner, however the LLM-Pruner uses a different heuristic for calculating weight importance: they do a first-order approximation of the loss, using the gradient of the loss with respect to each parameter. It is therefore unclear how $f_{C_{j}}(x)$ is calculated. Since this is a major part of the method, this would need to be explained very carefully in the final version of the paper.
>
> Thank you for the helpful comment. We agree that our initial phrasing was ambiguous. We have **revised Section 2.2.2** to remove the undefined term $f_{C_j}(x)$ and to explicitly adopt the first-order Taylor loss-sensitivity estimator—consistent with LLM-Pruner—computed per language.
>
> **What we changed:**
> We now explicitly define a loss-sensitivity estimator that is fully consistent with LLM-Pruner’s first-order approximation of the language-modeling loss. For each language $\ell$ and each coupled structure $\mathcal{C}_j$ constituent tensors \$\mathcal\{G\}(\mathcal\{C\}\_j)\$, we compute importance using a language-specific calibration subset \$\mathcal\{D\}\_\ell\$:
>
> **Per-parameter score:**
> \$\$
> s\_\ell(w) = \mathbb\{E\}\_{x \sim \mathcal\{D\}\_\ell} \left| \frac\{\partial L(x)\}\{\partial w\} \cdot w \right|
> \$\$
>
> **Per-tensor score:**
> \$\$
> s\_\ell(W) = \mathbb\{E\}\_{x \sim \mathcal\{D\}\_\ell} \left| \left\langle \frac\{\partial L(x)\}\{\partial W\}, W \right\rangle \right|
> \$\$
>
> **Structure-level aggregation (sum by default):**
> \$\$
> I\_\ell(\mathcal\{C\}\_j) = \sum\_{u \in \mathcal\{G\}(\mathcal\{C\}\_j)} s\_\ell(u)
> \$\$
>
> This revision removes the ambiguous term $f_{C_j}(x)$ and aligns our importance estimator with the standard first-order framework. Importantly, **this clarification does not change any empirical results**, and all trends and conclusions remain consistent with the original submission. We also include an ablation study on the importance-group aggregator hyperparameter in LLM-Pruner, with full details in Appendix A.2.
>
> ---
>
> > The advantage of Lang-Prune over LLM-Pruner after post-training is, though not negligible, hardly significant.
>
> We believe the reviewer is referring to **Figure 2**, where the performance gap between LLM-Pruner and Lang-Prune after post-training appears small. This is expected: due to the **limited post-training budget** (LoRA continued pre-training with 1B tokens), which reflects realistic constraints in low-resource multilingual pruning and recovery, the observable margin between approaches is inevitably modest.
>
> To further validate the conclusion, we **extended our analysis to additional benchmarks** to evaluate the post-training potential of Lang-Prune more comprehensively.
>
> Across translated-HellaSwag (shown previously) and three additional multilingual benchmarks, the results consistently show:
>
> 1. Mixed-data pruning substantially degrades multilingual QA performance, confirming that indiscriminate data mixing introduces cross-lingual interference.
> 2. Lang-Prune outperforms mixed-data pruning on most benchmarks—especially after light post-training—indicating better preservation of transferable multilingual features.
> 3. Post-training recovery is task- and language-dependent, with Lang-Prune benefiting more in settings with higher cross-lingual interference.
>
> **Conclusion:** These additional benchmarks confirm that Lang-Prune preserves cross-lingual generalization more effectively than mixed-data pruning and benefits more from light post-training. We appreciate the reviewer’s suggestion and have incorporated these results into the revised version.
>
> | Benchmark Accuracy $\uparrow$ | Belebele | translated-ARC | translated-HellaSwag | Global-MMLU |
> | ----------------------------- | -------- | -------------- | -------------------- | ----------- |
> | Qwen3-8B                      | 0.8796   | 0.5499         | 0.4691               | 0.6777      |
> | Mixed-data                    | 0.2331   | 0.2132         | 0.2523               | 0.2617      |
> | Mixed-data + training         | 0.2664   | 0.2311         | 0.2846               | 0.2695      |
> | Lang-Prune                    | 0.2626   | 0.2169         | 0.2639               | 0.2716      |
> | Lang-Prune + training         | 0.2686   | 0.2468         | 0.2904               | 0.2646      |

---

> ### Author Response · Authors · 2025-11-21
>
> ## Weaknesses
>
> > The authors should compare their work against other baselines; for example:
> >
> > - Multilingual Brain Surgeon (LREC 2024)
> >
> > - Pruning Multilingual Large Language Models for Multilingual Inference (EMNLP 2024)
>
> We thank the reviewer for the valuable pointers. Below, we clarify the differences between our method and the suggested baselines, and we additionally provide a new MBS-style comparison.
>
> - **Multilingual Brain Surgeon (MBS, LREC 2024)**
>
>   MBS focuses on *calibration data sampling* (e.g., balanced language mixtures) while keeping a single shared pruning criterion for all languages. It does **not** modify how multilingual importance is estimated or aggregated. In other words, it mitigates data imbalance but does not address cross-lingual interference arising from parameter sharing.
>
> - **Kim et al. (EMNLP 2024)**
>
>   This approach uses *alignment/translation-style supervision* to guide pruning for multilingual inference. It requires bilingual demonstrations and targets a different setting from our *one-shot structured pruning for multilingual language modeling*. While complementary, it cannot be directly applied in our scenario without additional supervised signals.
>
> - **Lang-Prune (ours)**
>
>   Lang-Prune computes *per-language* importance scores and aggregates them using a fairness-aware Max rule designed specifically to reduce cross-lingual interference during pruning. This directly addresses the multilingual coupling challenge that MBS and Kim et al. do not target.
>
> Kim et al.’s alignment-driven objectives operate in a different regime and could be integrated as an additional per-language signal when bilingual supervision exists. We approximate MBS-style calibration using publicly described mixing ratios; exact recipes are not available for the Qwen and Aya models. The results below show that **rebalancing calibration data (MBS-style) does help compared to uniform mixtures**, but **Lang-Prune achieves substantially larger improvements** due to its language-aware scoring. Further **discussion is provided in Appendix A.3**. We have also **added a discussion of these related works** in the Related Work section of the revised paper. We thank the reviewer for highlighting these references, which helped us clarify the distinctions and contributions of Lang-Prune.
>
> | PPL$\downarrow$ | EN:Others 3:1 | EN:Others 2:1 | EN:Others 5:1 | EN:ZH:ES:Others 3:2:2:1 | EN:ZH:ES:Others 4:2:2:1 | Uniform | Lang-Prune |
> | --- | --- | --- | --- | --- | --- | --- | --- |
> | ar | 506.51 | 110.02 | 298.05 | 313.12 | 331.37 | 208.99 | **22.87** |
> | cs | 211.97 | 80.93 | 386.84 | 182.11 | 159.93 | 102.79 | **14.57** |
> | de | 515.74 | 157.60 | 644.36 | 405.21 | 291.29 | 207.90 | **19.62** |
> | en | 1055.65 | 364.47 | 829.81 | 976.80 | 753.70 | 565.88 | **44.50** |
> | es | 379.91 | 143.36 | 411.79 | 326.87 | 289.16 | 205.65 | **22.02** |
> | id | 366.43 | 100.47 | 229.31 | 270.06 | 230.32 | 144.37 | **14.65** |
> | iw | 1226.97 | 539.44 | 3167.02 | 1791.35 | 1333.17 | 723.06 | **44.87** |
> | ru | 179.46 | 51.07 | 168.75 | 152.53 | 106.46 | 61.84 | **13.25** |
> | zh | 722.71 | 617.56 | 1318.28 | 629.44 | 528.75 | 575.77 | **22.35** |
> | avg | 573.93 | 240.55 | 828.24 | 560.83 | 447.13 | 310.69 | **24.30** |

---

> ### Author Response · Authors · 2025-11-21
>
> ## Weaknesses
>
> > There is little theoretical justification for the proposed method; to compensate for the relative lack of theoretical grounding, we would like to see more experimental studies. For example, we would like the authors to share insights on the language-specific importance of some neurons, with carefully curated samples to show how they trigger differently depending on the language.
>
> We appreciate the reviewer’s request for more fine-grained, language-level interpretability. To address this, we provide insights from our quantitative analysis and introduce a **new negative control discussion** to directly illustrate the "language-specific triggering" behavior.
>
> **1. Quantifying Language-Specific Specialization:**
> Our analysis of the **Recall Ratio** (Table 3) directly quantifies how well different methods align with language-specific triggers. We identified the top-p% most critical coupled structures for each language on the unpruned model and tracked their retention:
> * **Insight:** The results show that **Mixed-data pruning** fails to align with language-specific triggers, retaining only **~66%** of these critical structures.
> * **Lang-Prune Advantage:** In contrast, Lang-Prune retains **~81%**. This substantial gap empirically proves that distinct languages rely on distinct parameter groups. Lang-Prune succeeds precisely because it respects these non-overlapping activation patterns, whereas mixed calibration dilutes them.
>
> **2. Negative Control with Min Aggregator (New Discussion in Section 3.1):**
> To further justify our approach and validate the necessity of protecting language-specific neurons, we conducted a **negative control experiment** using a **Min aggregator**. This strategy selects the *minimum* importance score across languages, effectively targeting only the intersection (universally shared) neurons.
>
> * **Result:** We observed that this strategy maintains robustness for English but causes severe degradation for other languages.
> * **Conclusion:** This discovery indicates that even ostensibly "language-agnostic" (shared) neurons carry residual bias toward the dominant language. It provides strong empirical justification for our **Max aggregation** strategy: to ensure fair multilingual coverage, the pruning metric must explicitly protect the *union* of all language-specific structures.

---

> ### Author Response · Authors · 2025-11-21
>
> ## Questions
>
> > In Table 1, you quote perplexity after pruning in the multi-lingual and mono-lingual cases, and calculate a ratio. Would it not be more significant to compute the ratio of the *increase* in complexity over the base (non-pruned) model? I think it would be informative to show the perplexity of the base model in that table too (I am aware this is already shown in table 2).
>
> Thank you for the helpful suggestion. Table 1 is designed to isolate *cross-lingual interference* by comparing multilingual pruning against monolingual pruning. The IF ratio therefore quantifies degradation caused **solely** by multilingual calibration, independent of the base model.
>
> We agree that including the base model’s perplexity improves clarity. In the revised version, we added a **“Base Model” row** to Table 1 so readers can relate absolute PPL values to the IF ratios. We emphasize that while comparing to the base model measures *pruning difficulty*, the IF ratio specifically captures *cross-lingual interference*, which is the focus of this analysis.
>
> **Updated Table 1 (LLM-Pruner at 70% sparsity):**
>
> | PPL ↓           | ar        | cs        | de        | en        | es        | id        | iw        | ru        | zh        | avg       |
> | --------------- | --------- | --------- | --------- | --------- | --------- | --------- | --------- | --------- | --------- | --------- |
> | **Monolingual** | 42.71     | 81.99     | 82.82     | 236.44    | 85.19     | 77.52     | 46.87     | 57.73     | 36.47     | 83.08     |
> | **Mixed-data**  | 80.99     | 164.12    | 245.65    | 378.39    | 227.26    | 173.63    | 71.97     | 127.51    | 226.88    | 188.49    |
> | **IF score**    | **1.90×** | **2.00×** | **2.97×** | **1.60×** | **2.67×** | **2.24×** | **1.54×** | **2.21×** | **6.22×** | **2.27×** |
> | **Base Model**  | 8.10      | 10.46     | 10.66     | 10.50     | 9.58      | 12.77     | 11.19     | 11.13     | 10.27     | 10.52     |
>
> ---
>
> > How do you explain that Lang-Prune achieves better *average* perplexity than LLM-Pruner over multiple languages? One might assume that the compromise of using the maximum of per-language importance would, on average, lead to inferior results?
>
> This is an excellent question. While max aggregation may appear conservative, Lang-Prune outperforms LLM-Pruner because **mixed-language calibration often mis-ranks neuron importance**. With mean aggregation, a neuron critical for one language but unimportant for another receives a mid-range score; at high sparsity, many such neurons fall below the pruning threshold, harming the languages that rely on them. LLM-Pruner’s direct data-mixing strategy exhibits a similar effect: multilingual calibration blends heterogeneous activation patterns, introducing strong cross-lingual interference and consistent performance drops across languages (Table 1, Table 2).
>
> In contrast, **max aggregation explicitly preserves all language-relevant neuron groups**, preventing the dilution of language-specific signals under mixed-language scoring. This mitigates cross-lingual interference (Figure 1) and ensures that critical structures for each language are retained. Consequently, Lang-Prune with max aggregation achieves substantially better multilingual perplexity—even surpassing monolingual pruning at high sparsity—not by keeping more parameters, but by **keeping the right ones**. These insights are now discussed in the revised Section 3.1 ("Results Analysis") and Section 3.2 ("Analysis of Pruning Neuron Groups").

---

> ### Author Response · Authors · 2025-11-21
>
> ## Questions
>
> > How does language-aware pruning fundamentally differ from task-aware pruning?
>
> Thank you for the question. **In short:** multilingual pruning is fundamentally more structured and separable than multi-task (or multi-domain) pruning, which explains why Lang-Prune is effective for languages but not directly transferable to general task-aware settings.
>
> While the high-level idea of language-aware pruning resembles task-aware pruning, the two settings differ in **structural separability**. **The gap between human languages is substantially larger than the gap between closely related tasks**, especially in domains such as mathematics, computer science, or law, where tasks share vocabulary, reasoning patterns, and latent representations. Prior works show that multilingual models organize **discrete, language-specific neuron groups** associated with script, morphology, and linguistic form, and these clusters are far more separable than the subtle activation differences among closely related tasks [1,2,3]. This strong structural separation makes multilingual pruning more amenable to stable, structure-preserving strategies such as Lang-Prune.
>
> To validate this, we applied the same *per-domain importance + max aggregation* strategy to a **multi-domain setting** using Qwen3-8B, treating MMLU subfields (Law, Medical, Finance) as domains. Using their train/dev splits for calibration and the test split for evaluation at 50% sparsity, we observed that domain-aware pruning provides only **marginal improvements** over mixed-data pruning and remains far below the base model:
>
> | Avg Acc $\uparrow$ (stderr)  | Law                 | Medical              | Finance              |
> | ------------------------------ | ------------------- | -------------------- | --------------------- |
> | Base                           | $73.1\\% \pm 13.8\\%$ | $76.4\\% \pm 8.3\\%$   | $78.9\\% \pm 12.7\\%$   |
> | Mixed-data                     | $37.6\\% \pm 6.6\\%$  | $35.3\\% \pm 6.7\\%$   | $37.5\\% \pm 5.3\\%$    |
> | Lang-Prune (ours)              | $40.1\\% \pm 9.4\\%$  | $37.7\\% \pm 7.7\\%$   | $37.8\\% \pm 7.1\\%$    |
>
> Unlike the multilingual case, domain differences are **not distinct enough** to produce stable per-domain importance rankings, resulting in limited or inconsistent benefits. In contrast, language-level differences create strong, clean separations that Lang-Prune can reliably exploit. See details in Appendix A.4.
>
> References:
> - [1] Tang, Tianyi, et al. "Language-Specific Neurons: The Key to Multilingual Capabilities in Large Language Models." *ACL (1)*. 2024.
> - [2] Wang, Weixuan, et al. "Sharing matters: Analysing neurons across languages and tasks in llms." *arXiv preprint arXiv:2406.09265* (2024).
> - [3] Tan, Shaomu, Di Wu, and Christof Monz. "Neuron Specialization: Leveraging Intrinsic Task Modularity for Multilingual Machine Translation." *Proceedings of the 2024 Conference on Empirical Methods in Natural Language Processing*. 2024.

---

> ### Author Response · Authors · 2025-11-27
>
> Thank you for the constructive feedback and for engaging with our submission.  During the discussion period, we have incorporated several **key updates**:
>
> - **Clarified** the importance estimator and removed the ambiguous term in Eq. (1).
> - **Added** additional multilingual benchmarks and ablation studies.
> - **Included** comparisons and discussion of the suggested baselines.
> - **Expanded** analysis on language-specific importance and neuron behavior.
> - **Updated** tables and explanations to improve clarity of cross-lingual effects.
>
> We hope these revisions address the reviewer’s concerns. As the discussion window is short, **any further comments or questions would be greatly appreciated**, so we can ensure the strongest final version. We sincerely appreciate your engagement and constructive input, which have helped us significantly strengthen the paper.

---

### Official Review · Reviewer_nLNR · 2025-11-12

**Soundness:** 3
**Presentation:** 3
**Contribution:** 3
**Rating:** 6
**Confidence:** 3

**Summary:**

The authors propose Lang-Prune, a language-aware structured pruning framework built on top of LLM-Pruner. Lang-Prune computes per-language importance scores on small calibration sets and max-aggregates them to preserve any structure important for at least one language. It maintains full compatibility with LLM-Pruner coupled-structure mechanism and pruning schedules, introducing only minor changes to the scoring process. Experiment results show Lang-Prune indeed helps efficient and interference-free multi-lingual LLMs.

**Strengths:**

The method proposed in the paper is simple yet effective; the paper is easy to read overall.

**Weaknesses:**

Experiments in the paper are mostly limited to PPLs. I hope to see more benchmarks related to multilingual reasoning and generation, e.g. translation.

**Questions:**

1. Will the method work on larger-scale LLMs, e.g. 70B? (I understand that possibly the experiments could not be done due to hardware constraints.)
2. Only max/min/mean aggregators are used. What about other measures for aggregation, e.g. weighted aggregation?

---

> ### Author Response · Authors · 2025-11-21
>
> ## Weaknesses
>
> > Experiments in the paper are mostly limited to PPLs. I hope to see more benchmarks related to multilingual reasoning and generation, e.g. translation.
>
> We thank the reviewer for the suggestion to include additional benchmarks. To provide a more comprehensive evaluation, we extended our analysis beyond *translated-HellaSwag* to three diverse multilingual benchmarks: *Belebele* (reading comprehension), *translated-ARC* (grade-school reasoning), and *Global-MMLU* (broad multilingual knowledge). The results show:
>
> 1. **Mixed-data pruning** substantially degrades multilingual performance across all benchmarks, confirming that indiscriminate mixing of calibration data introduces cross-lingual interference.
> 2. **Lang-Prune** consistently outperforms mixed-data pruning, especially after light post-training, preserving transferable multilingual features more effectively.
> 3. **Post-training recovery** varies by task and language, with Lang-Prune showing larger gains where cross-lingual interference is higher.
>
> **Conclusion:** These additional benchmarks confirm that Lang-Prune better preserves cross-lingual generalization and benefits more from light post-training compared to mixed-data pruning. We have incorporated these results in the revised version to strengthen the empirical evaluation (Appendix A.6).
>
> | Benchmark Accuracy $\uparrow$ | Belebele | translated-ARC | translated-HellaSwag | Global-MMLU |
> | ----------------------------- | -------- | -------------- | -------------------- | ----------- |
> | Qwen3-8B                      | 0.8796   | 0.5499         | 0.4691               | 0.6777      |
> | Mixed-data                    | 0.2331   | 0.2132         | 0.2523               | 0.2617      |
> | Mixed-data + training         | 0.2664   | 0.2311         | 0.2846               | 0.2695      |
> | Lang-Prune                    | 0.2626   | 0.2169         | 0.2639               | 0.2716      |
> | Lang-Prune + training         | 0.2686   | 0.2468         | 0.2904               | 0.2646      |

---

> ### Author Response · Authors · 2025-11-21
>
> ## Questions
>
> > Will the method work on larger-scale LLMs, e.g. 70B? (I understand that possibly the experiments could not be done due to hardware constraints.)
>
> We appreciate the reviewer’s understanding regarding hardware constraints. While we could not perform pruning and evaluation on 70B models due to compute limits, we conducted a comprehensive scaling analysis on the **Qwen3 family ranging from 0.6B to 14B parameters** to verify the trend:
>
> | Model Size                 | 0.6B      | 1.7B      | 4B        | 8B        | 14B       |
> | -------------------------- | --------- | --------- | --------- | --------- | --------- |
> | Base Model                 | 23.95     | 16.52     | 14.28     | 11.02     | 9.50      |
> | Mixed-data  (50%)          | 52.38     | 80.96     | 40.72     | 310.69    | 555.36    |
> | Monolingual  (50%)         | **47.59** | **36.47** | 32.72     | 36.72     | 69.10     |
> | **Lang-Prune (ours, 50%)** | 56.07     | 45.68     | **31.43** | **24.30** | **31.60** |
>
> **Key Findings:**
> 1.  **Clear Advantage at Scale (4B+):** Lang-Prune significantly outperforms both Mixed-data and Monolingual baselines on 4B, 8B, and 14B models. The gap widens as model size increases (e.g., at 14B, Lang-Prune is ~2x better than Monolingual).
> 2.  **Small Model Sensitivity:** For tiny models (0.6B, 1.7B), Lang-Prune and Mixed-data perform worse than Monolingual pruning. We attribute this to the lack of structural redundancy in small models; explicitly reserving capacity for all languages via the `Max` aggregator becomes too aggressive when the parameter budget is extremely tight.
>
> **Conclusion:** The results strongly suggest that Lang-Prune's benefits amplify with model scale. Given that 70B models possess even greater redundancy and multilingual capacity than 14B models, we expect Lang-Prune to be highly effective in that regime. Details are provided in Appendix A.8.
>
> ---
>
> > Only max/min/mean aggregators are used. What about other measures for aggregation, e.g. weighted aggregation?
>
> This is an excellent question. While we focused on `Max` for its parameter-free property and "fairness" (protecting any language-critical structure), we **did explore weighted strategies** in our ablation studies (Appendix A.3, Tables 9).
>
> 1.  **Weighted Aggregation Performance:** We simulated weighted aggregation by adjusting the calibration data mixture ratios (e.g., English:Others = 3:1, or English:Chinese:Spanish:Others = 4:2:2:1). Since importance scores are derived from calibration loss, weighting the data distribution is a proxy for weighted score aggregation.
>     * **Result:** As shown in Appendix Table 9, even the best weighted mixture (PPL 240.55) performed significantly worse than Lang-Prune's `Max` aggregator (PPL 24.30) on Qwen3-8B.
>     * **Reason:** Weighted aggregation implicitly creates a trade-off, sacrificing structures important to low-weight languages to favor high-weight ones. In contrast, `Max` acts as a "union" operation, preserving structures if they are critical to *at least one* language, which effectively mitigates cross-lingual interference without manual tuning.
>
> 2.  **Why Max/Mean/Min:**
>     * **Max:** Implements a worst-case protection strategy (our proposed solution).
>     * **Mean:** Represents the standard average-case handling.
>     * **Min:** Serves as a negative control.
>
> We have clarified in the revised paper that while weighted aggregation is possible, our empirical results suggest that the parameter-free `Max` strategy is more robust for maintaining multilingual coverage than manually tuning language weights.

---

> > ### Comment · Reviewer_nLNR · 2025-11-21
> > **Response from Reviewer nLNR**
> >
> > Thank you very much for your response. I appreciate the authors' efforts adding benchmarks; I am also satisfied with the trend when the model is scaled-up. Thus, I will maintain my positive rating, leaning towards acceptance.

---

> > > ### Author Response · Authors · 2025-11-21
> > >
> > > Thank you for your follow-up and for taking the time to re-evaluate our revision. We appreciate your positive assessment of the newly added benchmarks and the clarification regarding scaling trends. We are grateful that you are maintaining your positive rating and leaning toward acceptance.
> > >
> > > Please let us know if any further clarification would be helpful—we are happy to provide additional details.

---

### Author Response · Authors · 2025-11-24

# Summary of Revisions and Responses
We sincerely thank the reviewers for their thoughtful and constructive feedback. We are encouraged by the recognition of Lang-Prune’s core contributions, including its simplicity, effectiveness, and cross-lingual generalization:

* **Simplicity & Effectiveness:** Reviewers highlighted Lang-Prune as a “simple yet effective” (*Reviewer nLNR*) and “straightforward, plug-and-play” solution that “works surprisingly well” with minimal computational overhead (*Reviewer haRh*, *Reviewer c7K3*).
* **Conceptual Insight:** We appreciate the note that our method “effectively identifies a key limitation” in prior multilingual pruning (*Reviewer c7K3*) and provides “interesting conceptual insights” into cross-lingual interference (*Reviewer haRh*).
* **Robustness & Generalization:** Reviewers highlighted the “meaningful ablations” (*Reviewer haRh*) and “consistently superior performance” with strong generalization across diverse languages (*Reviewer c7K3*, *Reviewer gU8x*).

The reviewers’ constructive suggestions motivated several key improvements, summarized below:

1. **Clarified Theoretical Definition (Section 2.2.2):** We have revised the importance estimation formula to remove ambiguities and make the definition precise.
2. **Enhanced Related Work & Baselines (Appendix A.3):** We expanded comparisons with MBS-style calibration and alignment-oriented pruning to better contextualize our contribution and highlight transferability.
3. **Clarified Scope vs. Multi-Domain Pruning (Appendix A.4):** Experiments on multi-domain pruning (Law/Medical/Finance) show that Lang-Prune is more effective for language-specific tasks due to strong structural separability, distinct from general domain tasks.
4. **Validated Robustness to Calibration Size (Appendix A.5):** Ablations across calibration dataset sizes show that Lang-Prune consistently outperforms baselines. We also provide an analysis of why standard monolingual pruning can degrade at high sparsity with larger calibration datasets.
5. **Broadened Post-Training Benchmarking (Appendix A.6):** Additional evaluations on three diverse multilingual benchmarks indicate that Lang-Prune maintains strong cross-lingual generalization and adaptation compared to mixed-data pruning.
6. **Tested Unstructured Pruning (Appendix A.7):** Applying Lang-Prune within Wanda suggests that effective multilingual pruning benefits from preserving semantic structures (units) rather than isolated weights.
7. **Expanded Model-Scale Evaluation (Appendix A.8):** We extended our evaluation to the Qwen3 family (0.6B–14B parameters), observing larger gains at larger model scales consistent with increased structural redundancy.

These revisions strengthen the manuscript’s clarity, theoretical rigor, and empirical validation. **We hope these revisions address the reviewers’ concerns and welcome any further questions or suggestions.**

---

### Meta-Review · Area_Chair_SSux · 2026-01-05

**Summary:**

This paper focuses on pruning of multilingual LLM-s. The authors make the observation that when pruning a multi-lingual LLM with mixed-language calibration, the model performance on certain languages might experience a significant deterioration,  a phenomenon the authors term cross-lingual interference. The authors then introduce LangPrune, a language-aware modification to structured pruning that calculates per-language importance on small calibration datasets and uses max-aggregation to preserve units that are critical to any of languages. The authors conduct extensive evaluations across nine languages and observe that LangPrune consistently improves both average and worst-case performance compared to the baseline, LLM-Pruner.

**Reviewer Concerns:**

The reviewers expressed concerns about the narrow scope of the evaluation, with experiments focusing mostly on perplexity with limited evaluation on multilingual downstream tasks (reasoning, translation, etc), and minimal QA coverage (only HellaSwag).  Since the initial experiments were done on an 8b model, there were also questions about the scalability of LangPune to bigger models. A couple of reviewers noted that after post-training advantage of LangPrune over LLM-Pruner is marginal. Another issue was omission of relevant multilingual pruning baselines such as Multilingual Brain Surgeon, EMNLP 2024 methods. Furthermore, one reviewer asked whether the importance scoring approach would work for other (unstructured) pruning methods such as Wanda.

In the rebuttal the authors provided new results on three multilingual benchmarks: Belebele, translated-ARC, and Global-MMLU, both with and without additional training. While the authors cite the new results as evidence of LargePrune’s superiority over mixed data training, the difference are rather marginal, and seem to become even smaller after training (Table 13 in Appendix A6). In fact, for Global MMLU mixed data+training is marginally better than LangPrune+training. An additional limiting factor is that the results are provided only for the 70% sparsity level.

To analyze the scalability of the approach, the authors presented new results with Qwen3 family of models ranging from 0.6B to 14B parameters. The results show that Lang-Prune significantly outperforms Mixed-data and Monolingual baselines on larger models (4B and over). However, what is concerning is that with LangPrune, the perplexity increases when going from 8B to 14B. So the authors assertion that LangPrune will be effective of larger model sizes is not supported by the results -  they might be effective in compassion to the baselines, but over 20% increase in perplexity while almost doubling the model size raises questions about practical feasibility of LangPrune (or those baselines) for larger models.


The authors also provided new results on using LangPrune with Wanda, and observe that it performs poorly compared to baselines. The authors assert that this is because LangPrune’s reliance on consistent pruning units. However, this point needs more analysis and evaluation, e.g., trying LangPrune with other structural pruning methods. More importantly, qualitative comparison of Table 14 with Table 2 suggests that unstructured pruning(Wanda) + mixed data might yield a significantly better perplexity than structured pruning (LLMPruner) + LangPrune, and thus might be a more practical overall approach for real-world use cases.  Finally, the AC would like to note that the experimental focus on aggressive pruning (70% sparsity) is somewhat puzzling. Even though LangPruner is comparably better compared to baselines in this regime, all the approaches experience a drastic  performance drop (e.g., from 0.67 to 0.27 on Global MMLU) making them unusable for most real-world applications.

**Reviewer Scores:**

nLNR - 6 unchanged;
gU8x - 4 unchanged;
haRh - 8 unchanged;
c7K3 - 2 unchanged;

---

### Decision · Program_Chairs · 2026-01-26

Reject